# Intrinsic Dimension for Large-Scale Geometric Learning

**Maximilian Stubbemann**  *stubbemann@cs.uni-kassel.de*
*Knowledge & Data Engineering Group, University of Kassel, Kassel, Germany*

**Tom Hanika**  *tom.hanika@cs.uni-kassel.de*
*Knowledge & Data Engineering Group, University of Kassel, Kassel, Germany*

**Friedrich Martin Schneider**  *martin.schneider@math.tu-freiberg.de*
*Institute of Discrete Mathematics and Algebra, TU Bergakademie Freiberg, Freiberg, Germany*

**Reviewed on OpenReview:** *https: // openreview. net/ forum? id= 85BfDdYMBY*

## Abstract

The concept of dimension is essential to grasp the complexity of data. A naive approach to determine the dimension of a dataset is based on the number of attributes. More sophisticated methods derive a notion of intrinsic dimension (ID) that employs more complex feature functions, e.g., distances between data points. Yet, many of these approaches are based on empirical observations, cannot cope with the geometric character of contemporary datasets, and do lack an axiomatic foundation. A different approach was proposed by V. Pestov, who links the intrinsic dimension axiomatically to the mathematical concentration of measure phenomenon. First methods to compute this and related notions for ID were computationally intractable for large-scale real-world datasets. In the present work, we derive a computationally feasible method for determining said axiomatic ID functions. Moreover, we demonstrate how the geometric properties of complex data are accounted for in our modeling. In particular, we propose a principle way to incorporate neighborhood information, as in graph data, into the ID. This allows for new insights into common graph learning procedures, which we illustrate by experiments on the Open Graph Benchmark.

## 1  Introduction

Contemporary real-world datasets employed in artificial intelligence are often large in size and comprised of complex structures, which distinguishes them from Euclidean data. To consider these properties appropriately is a challenging task for procedures that analyze or learn from said data. Moreover, with increasing complexity of real-world data, the necessity arises to quantify to which extent this data suffer from the curse of dimensionality. The common approach for estimating the dimension curse of a particular dataset is through the notion of *intrinsic dimension* (ID) (Bac & Zinovyev, 2020; Granata & Carnevale, 2016; Pestov, 2007). There exists a variety of work on how to estimate the ID of datasets (Facco et al., 2017; Levina & Bickel, 2004; Costa et al., 2005; Gomtsyan et al., 2019; Bac & Zinovyev, 2020). Most approaches to quantify the ID are based on distances between data points, assuming the data to be Euclidean. A multitude of works base their modeling on the manifold hypothesis (Cloninger & Klock, 2021; Gomtsyan et al., 2019), which assumes that the observed data is embedded in a manifold of low dimension (compared to the number of data attributes). The ID then is an approximation of the dimension of this manifold. Pestov (2000) proposed a different concept of intrinsic dimension by linking it to the mathematical concentration of measure phenomenon. His modeling is based on a thorough axiomatic approach (Pestov, 2007; 2008; 2010) which resulted in a novel class of intrinsic dimension functions. In contrast to the manifold hypothesis, Pestov's ID functions measure to which extent a dataset is affected by the curse of dimensionality, i.e., to which extent the complexity of the dataset hinders the discrimination of data points. Yet, to compute said ID functions is an intractable computational endeavor. This limitation was overcome in principle by an adaptation to *geometric datasets* (Hanika et al., 2022). However, two limitations persisted: First, the computational effort was

found to remain quadratic in the number of data points, which is insufficient for datasets of contemporary size; second, it is unclear how to account for complex structure, such as in graph data.

With this in mind, we propose in the present work a default approach for computing the intrinsic dimension of geometric data, such as graph data, as used in graph neural networks. To do this, we revisit the computation of the ID based on distance functions (Hanika et al., 2022) and overcome, in particular, the inherent computational limitations in the works by Pestov (2007) and Hanika et al. (2022). In detail, we derive a novel approximation formula and present an algorithm for its computation. This allows us to compute ID bounds for datasets that are magnitudes larger than in earlier works. That equipped with, we establish a natural approach to compute the ID of graph data.

We subsequently apply our method to seven real-world datasets and relate the obtained results to the observed performances of classification procedures. Thus, we demonstrate the practical computability of our approach. In addition, we study the extent to which the intrinsic dimension reveals insights into the performance of particularly classes of Graph Neural Networks. Our code is publicly available on GitHub.[1]

## 2 Related Work

In numerous works, the intrinsic dimension is estimated using the pairwise distances between data points (Chávez et al., 2001; Grassberger & Procaccia, 2004). More sophisticated approaches use distances to nearest neighbors (Facco et al., 2017; Levina & Bickel, 2004; Costa et al., 2005; Gomtsyan et al., 2019). All these works have in common, that they assume the data to be Euclidean and that they favor local properties.

Recent work has drawn different connections between intrinsic dimension (ID) and modern learning theory. For example, Cloninger & Klock (2021) show that functions of the form $f(x) = g(\phi(x))$, where $\phi$ maps into a manifold of lower dimension, can be approximated by neural networks. On the other hand, Wojtowytsch & E (2020) prove that modern artificial neural networks suffer from the curse of dimensionality in the sense that gradient training on high dimensional data may converge insufficiently. Additional to these theoretical results, there is an increasing interest of empirically estimating the ID of contemporary learning architectures. Li et al. (2018) study the ID of neural networks by replacing high dimensional parameter vectors with lower dimensional ones. Their approach results in a non-deterministic ID. More recent works studied ID in the realm of geometric data and their standard architectures. Ansuini et al. (2019) investigate the ID for convolutional neural networks (CNN). In detail, they are interested to which extent the ID changes at different hidden layers and how this is related to the overall classification performance. Another work (Pope et al., 2020) associates an ID to popular benchmark image datasets. These two works on ID estimators do solely rely on the metric information of the data and do not consider any geometric structure of image data.

Our approach allows to incorporate such underlying geometric structures while incorporating the mathematical phenomenon of measure concentration (Gromov & Milman, 1983; Milman, 1988; 2010). Linking this phenomenon to the occurrence of the dimension curse was done by Pestov (Pestov, 2000; 2007; 2008; 2010). He based his considerations on a thorough axiomatic approach using techniques from metric-measure spaces. The resulting ID functions unfortunately turn out to be practically incomputable. In contrast, Bac & Zinovyev (2020) investigate computationally feasible ID estimators that are related to the concentration phenomenon. Yet, their results elude a comparable axiomatic foundation. Our modeling for the ID of large and geometric data is based on Hanika et al. (2022). We build on their axiomatization and derive a computationally feasible method for the intrinsic dimension of large-scale geometric datasets.

## 3 Intrinsic Dimension

Since our work is based on the formalization from Hanika et al. (2022), we shortly revisit their modeling and recapitulate the most important structures. Based on this, we derive and prove an explicit formula to compute the ID for the special case of finite geometric datasets. This first result is essential for Section 4.

---

[1] https://github.com/mstubbemann/ID4GeoL

Let $\mathcal{D} = (X, F, \mu)$, where $X$ is a set and $F \subseteq \mathbb{R}^X$ is a set of functions from $X$ to $\mathbb{R}$, in the following called *feature functions*. We require that $\sup_{x,y \in X} d_F(x, y) < \infty$, where $d_F(x, y) := \sup_{f \in F} |f(x) - f(y)|$. If $(X, d_F)$ constitutes a complete and separable metric space such that $\mu$ is a Borel probability measure on $(X, d_F)$, we call $\mathcal{D}$ a *geometric dataset* (GD). The aforementioned properties are satisfied when it holds that $0 < |X|, |F| < \infty$ and that $F$ can discriminate all data points, i.e., $d_F(x, y) > 0$ for all $x, y \in X$ with $x \neq y$.

Two geometric datasets $\mathcal{D}_1 := (X_1, F_1, \mu_1), \mathcal{D}_2 := (X_2, F_2, \mu_2)$ are isomorphic if there exists a bijection $\phi : X_1 \to X_2$ such that $\overline{F_2} \circ \phi = \overline{F_1}$ and $\phi_*(\mu_1) = \mu_2$, where $\phi_*(\mu_1)(B) := \mu_1(\phi^{-1}(B))$ is the *push-forward measure* and the closures are taken with respect to point-wise convergence. From this point on we identify a geometric dataset with its isomorphism class. The triple $(\{\emptyset\}, \mathbb{R}, \nu_{\{\emptyset\}})$ represents the *trivial geometric dataset*.

Pestov (2007) defines the curse of dimensionality as "[...] a name given to the situation where all or some of the important features of a dataset sharply concentrate near their median (or mean) values and thus become non-discriminating. In such cases, X is perceived as intrinsically high-dimensional." Thus, the ID estimator aims to compute to which extent the features allow to discriminate different data points. For a specific feature $f \in F$, Hanika et al. (2022) therefore defines the *partial diameter* of $f$ with regard to a specific $\alpha \in (0, 1)$ such that it displays to which extent $f$ can discriminate subsets with minimal measure $1 - \alpha$, i.e., via

$$\text{PartDiam}(f_*(\nu), 1-\alpha) = \inf\{\text{diam}(B) \mid B \subseteq \mathbb{R} \text{ Borel}, \nu(f^{-1}(B)) \geq 1 - \alpha\},$$

where $\text{diam}(B) := \sup_{a,b \in B} |a - b|$. The *observable diameter* with respect to $\alpha$ then defines to which extent $F$ can discriminate points with minimum measure 1-$\alpha$ by being defined as the supremum of the partial diameter of all $f \in F$, i.e, $\text{ObsDiam}(\mathcal{D}, -\alpha) := \sup_{f \in F} \text{PartDiam}(f_*(\mu), 1 - \alpha)$. To observe the discriminability for different minimal measures $\alpha$, the *discriminability* $\Delta(\mathcal{D})$ of $\mathcal{D}$ and the *intrinsic dimension* $\partial(\mathcal{D})$ are defined via

$$\partial(\mathcal{D}) := \frac{1}{\Delta(\mathcal{D})^2}, \quad \text{where} \quad \Delta(\mathcal{D}) := \int_0^1 \text{ObsDiam}(\mathcal{D}, -\alpha) \, d\alpha. \tag{1}$$

In other words, lower values of intrinsic dimensionality correspond to geometric datasets with points that can be better discriminated by the given set of feature functions. This intrinsic dimension function is, in principle, applicable to a broad variety of geometric data, such as metric data, graphs or images. This applicability arises from the possibility to choose suitable feature functions which reflect the underlying data structure. The appropriate choice of feature functions is part of Section 5. Furthermore, the ID $\partial(\mathcal{D}) = \frac{1}{\Delta(\mathcal{D})^2}$ respects the formal axiomatization (Hanika et al., 2022) for ID functions, informally:

- **Axiom of concentration:** A sequence of geometric datasets converges against the constant dataset (meaning having no chance to separate data points!), if and only if their IDs diverge against infinity.

- **Axiom of feature antitonicity:** If dataset $\mathcal{D}_1$ has more feature functions then $\mathcal{D}_0$ (i.e. having potentially more information to separate data points), it should have a lower intrinsic dimension.

- **Axiom of continuity:** If a sequence of geometric datasets converge against a specific geometric dataset, the sequence of the IDs should converge against the ID of the limit geometric dataset.

- **Axiom of geometric order of divergence:** If a sequence of geometric datasets converges against the constant dataset, its IDs should diverge against infinity with the same order as $\frac{1}{\Delta((D))^2}$ does.

## 3.1 Intrinsic Dimension of Finite Data

We want to apply Equation (1) to real-world data. In the following, let $\mathcal{D} = (X, F, \nu)$ such that $0 < |X| < \infty$ and $0 < |F| < \infty$ and let $\nu$ be the normalized counting measure on $X$, i.e., $\nu(M) := \frac{|M|}{|X|}$ for $M \subseteq X$. In this case, it is possible to compute the partial diameter and Equation (1), as we show in the following. Let $\alpha \in (0, 1)$ and let $c_\alpha := \lceil |X|(1-\alpha) \rceil$. The following arguments were already hinted in previous work (Hanika et al., 2022), yet not formally discussed or proven.

**Lemma 3.1.** *For $f \in F$ it holds that*

$$\text{PartDiam}(f_*(\nu), 1 - \alpha) = \min_{|M| = c_\alpha} \max_{x,y \in M} |f(x) - f(y)|.$$

*Proof.* It holds that

$$\text{PartDiam}(f_*(\nu), 1-\alpha) = \inf\{\text{diam}(B) \mid B \subseteq \mathbb{R} \text{ Borel}, \nu(f^{-1}(B)) \geq 1 - \alpha\}$$
$$= \inf\{\text{diam}(B) \mid B \subseteq \mathbb{R} \text{ Borel}, |\{x \in X \mid f(x) \in B\}| \geq c_\alpha\}.$$

We have to show that

$$\inf\{\text{diam}(B) \mid B \subseteq \mathbb{R} \text{ Borel}, |\{x \in X \mid f(x) \in B\}| \geq c_\alpha\} =$$
$$\min\{\max_{x,y \in M} |f(x) - f(y)| \mid M \subseteq X, |M| \geq c_\alpha\}. \tag{2}$$

"$\leq$:" We show that $\{\text{diam}(B) \mid B \subseteq \mathbb{R} \text{ Borel}, |\{x \in X \mid f(x) \in B\}| \geq c_\alpha\} \supseteq \{\max_{x,y \in M} |f(x) - f(y)| \mid M \subseteq X, |M| \geq c_\alpha\}$. Let $z := \max_{x,y \in M} |f(x) - f(y)|$ such that $M \subseteq X$ with $|M| \geq c_\alpha$. Without loss of generality we assume that

$$\forall x \in X : (\exists m_1, m_2 \in M : f(m_1) \leq f(x) \leq f(m_2) \implies x \in M).$$

Let $b := \max_{x \in M} f(x)$, $a := \min_{x \in M} f(x)$, then $M = \{x \in X \mid f(x) \in [a,b]\}$. Hence, $z = b - a \in \{\text{diam}(B) | B \subseteq \mathbb{R} \text{ Borel}, |\{x \in X | f(x) \in B\}| \geq c_\alpha\}$.

"$\geq$:" Let $B \subseteq \mathbb{R}$ be Borel with $|\{x \in X \mid f(x) \in B\}| \geq c_\alpha$. Furthermore, let $M := \{x \in X \mid f(x) \in B\}$. It holds that $\text{diam}(B) = \sup_{x,y \in B} |x - y| \geq \max_{x,y \in M} |f(x) - f(y)|$ because of the choice of $M$. As $B$ was chosen arbitrarily, it follows "$\geq$".

Finally, we need that

$$\min\{\max_{x,y \in M} |f(x) - f(y)| \mid |M| \geq c_\alpha\} = \min\{\max_{x,y \in M} |f(x) - f(y)| \mid |M| = c_\alpha\}.$$

"$\leq$" follows directly from the fact that $\{\max_{x,y \in M} |f(x) - f(y)| \mid |M| \geq c_\alpha\} \supseteq \{\max_{x,y \in M} |f(x) - f(y)| \mid |M| = c_\alpha\}$. "$\geq$" follows from the fact that for every $|M| \geq c_\alpha$ and for every $N \subseteq M$ with $|N| = c_\alpha$ the following equation holds: $\sup_{x,y \in M} |f(x) - f(y)| \geq \sup_{x,y \in N} |f(x) - f(y)|$. $\square$

This lemma allows for a more tractable formula for the computation of the partial diameter of a finite GD. That in turn enables the following theorem.

**Theorem 3.2.** *It holds that*

$$\Delta(\mathcal{D}) = \frac{1}{|X|} \sum_{k=2}^{|X|} \max_{f \in F} \min_{\substack{M \subseteq X \\ |M|=k}} \max_{x,y \in M} |f(x) - f(y)|. \tag{3}$$

*Proof.* Let $g : (0,1) \to \mathbb{R}, \alpha \mapsto \max_{f \in F} \min_{M \subseteq X, |M|=c_\alpha} \max_{x,y \in M} |f(x) - f(y)|$. Because of Lemma 3.1 we know that $\Delta(\mathcal{D}) = \int_0^1 g(\alpha) \, d\alpha$. The function $g$ is a step function which can be expressed for each $\alpha \in (0,1)$ via

$$g(\alpha) = \sum_{k=1}^{|X|} \mathbb{1}_{\left(\frac{|X|-k}{|X|}, \frac{|X|+1-k}{|X|}\right)}(\alpha) \max_{f \in F} \min_{\substack{M \subseteq X \\ |M|=k}} \max_{x,y \in M} |f(x) - f(y)|$$

almost everywhere. Hence, Equation (3) follows from the definition of the Lebesgue-Integral with the fact that $\min_{M \subseteq X, |M|=1} \max_{x,y \in M} |f(x) - f(y)| = 0$. $\square$

In general, the addition of features should lower the ID since we have additional information that helps to discriminate the data. However, there are certain features that are not helping to further discriminate data points. These are for example:

1. Constant features. This is due to the fact that for a constant feature $f$ it always holds for all $M \subseteq X$ that $\max_{x,y \in M} |f(x) - f(y)| = 0$.

2. Permutations of already existing features. Let $\tilde{f} : X \to \mathbb{R}$ have the form $\tilde{f} = f \circ \pi$ with $\pi : X \to X$ being a permutation on $X$. Then there exist for all $M \subseteq X$ with $|M| = k$ a set $N \subseteq X$ with $|N| = k$ and $\max_{x,y \in M} |f(x) - f(y)| = \max_{x,y \in N} |\tilde{f}(x) - \tilde{f}(y)|$ and vice versa.

Thus, we have the following Lemma.

**Lemma 3.3.** *Let $\mathcal{D} = (X, F, \mu)$ be a finite geometric dataset. Furthermore, let $\hat{F}$ be a set of constant functions $X \to \mathbb{R}$ and let $\tilde{F}$ be a set of functions $X \to \mathbb{R}$ such that there exist for each $\tilde{f} \in \tilde{F}$ a $f \in F$ and a permutation $\pi : X \to X$ with $\tilde{f} = f \circ \pi$. Let $\mathcal{E} := (X, F \cup \hat{F} \cup \tilde{F}, \mu)$. Then it holds that $\partial(D) = \partial(E)$.*

### 3.2 Computing the Intrinsic Dimension of Finite Data

In this section we will propose an algorithm for computing the ID based on Equation (3). For this, given a finite geometric dataset $\mathcal{D}$, we use the shortened notations $\phi_{k,f}(\mathcal{D}) := \min_{M \subseteq X, |M|=k} \max_{x,y \in M} |f(x) - f(y)|$ and $\phi_k(\mathcal{D}) := \max_{f \in F} \phi_{k,f}(\mathcal{D})$. Then, Equation (3) can be written as

$$\Delta(\mathcal{D}) = \frac{1}{|X|} \sum_{k=2}^{|X|} \phi_k(\mathcal{D}) = \frac{1}{|X|} \sum_{k=2}^{|X|} \max_{f \in F} \phi_{k,f}(\mathcal{D}). \tag{4}$$

The straightforward computation of Equation (4) is hindered by the task to iterate through all subsets $M \subseteq X$ of size $k$. This yields an exponential complexity with respect to $|X|$ for computing $\Delta(\mathcal{D})$. We can overcome this towards a quadratic computational complexity in $|X|$ using the following concept.

**Definition 3.4.** *(Feature Sequence) For a feature $f \in F$ let $l_{f,\mathcal{D}} \in \mathbb{R}^{|X|}$ be the increasing sequence of all values $f(x)$ for $x \in X$. We call $l_{f,\mathcal{D}} = (l_1^{f,\mathcal{D}}, \dots, l_{|X|}^{f,\mathcal{D}})$ the* feature sequence *of $f$.*

Using these sequences, the following lemma allows us to efficiently compute $\phi_{k,f}(\mathcal{D})$.

**Lemma 3.5.** *For $k \in \{2, \dots, |X|\}, f \in F$ and $l_{f,\mathcal{D}}$, it holds that*

$$\phi_{k,f}(\mathcal{D}) = \min\{l_{k+j}^{f,\mathcal{D}} - l_{1+j}^{f,\mathcal{D}} \mid j \in \{0, \dots, |X| - k\}\}.$$

*Proof.* For all $j \in \{0, \dots, |X|-k\}$ there exist $M \subseteq X$ with $|M| = k$ and $l_{k+j}^{f,\mathcal{D}} - l_{1+j}^{f,\mathcal{D}} = \max_{x,y \in M} |f(x) - f(y)|$. Thus, it is sufficient to show $\phi_{k,f}(\mathcal{D}) \in \{l_{k+j}^{f,\mathcal{D}} - l_{1+j}^{f,\mathcal{D}} \mid j \in \{0, \dots, |X|-k\}\}$. Choose $M \subseteq X$ with $|M| = k$ such that $\phi_{k,f}(\mathcal{D}) = \max_{x,y \in M} |f(x) - f(y)|$ holds. Furthermore, let $l^M := (l_1^M, \dots, l_k^M)$ be the increasing sequence of values $f(m)$ for $m \in M$ and let $j \in \{0, \dots, |X| - k\}$ such that $l_1^M = l_{1+j}^{f,\mathcal{D}}$. Since $l^M$ is an ordered sequence of which each element is also an element of the ordered sequence $l_{f,\mathcal{D}}$, it holds that $l_k^M \geq l_{k+j}^{f,\mathcal{D}}$ and thus $l_k^M - l_1^M \geq l_{j+k}^{f,\mathcal{D}} - l_{j+1}^{f,\mathcal{D}}$. There is an $N \subseteq X$ with size $k$ such that $\max_{x,y \in N} |f(x) - f(y)| = l_{k+j}^{f,\mathcal{D}} - l_{k+1}^{f,\mathcal{D}}$. Since $M \subseteq X$ is of size $k$ such that $\max_{x,y \in M} |f(x) - f(y)|$ is minimal, it follows $l_k^M - l_1^M = \max_{x,y \in M} |f(x) - f(y)| \leq \max_{x,y \in N} |f(x) - f(y)| = l_{k+j}^{f,\mathcal{D}} - l_{k+1}^{f,\mathcal{D}}$, hence $\phi_{k,f}(\mathcal{D}) = \max_{x,y \in M} |f(x) - f(y)| = l_k^M - l_1^M = l_{k+j}^{f,\mathcal{D}} - l_{k+1}^{f,\mathcal{D}}$. $\square$

To sum up, Lemma 3.5 enables the efficient computation of $\phi_{k,f}(\mathcal{D})$ via a sliding window, i.e., by using only pairs of elements $(l_{1+j}^{f,\mathcal{D}}, l_{k+j}^{f,\mathcal{D}})$. The algorithm based on this is shown in Algorithm 1. We want to provide a brief description of the most relevant steps. In Line 4 we iterate through the sizes of $X$ by setting $k \in \{2, \dots, |X|\}$ in order to compute $\phi_k(\mathcal{D})$ in Lines 6 and 7. For this we also need to iterate over all $f \in F$ (Line 5) to compute the necessary values of $\phi_{k,f}(\mathcal{D})$ in Line 6. For a given $f \in F, k \in \{1, \dots, |X|\}$, Line 6 consumes $|X| - k + 1$ subtraction operations. Assuming that computing feature values can be done in constant time, the runtime for computing $\Delta(\mathcal{D})$ from the feature sequences is $\mathcal{O}(|F| \sum_{k=2}^{|X|} |X| - k + 1) = \mathcal{O}(|F| \sum_{k=1}^{|X|-1} k) = \mathcal{O}(|F||X|^2)$. The creation of all feature sequences requires $\mathcal{O}(|F||X| \log(|X|))$ computations , which is negligible compared to the aforementioned complexity. Thus, Algorithm 1 has quadratic complexity with respect to $|X|$. Therefore, Algorithm 1 is a straightforward and easy to implement

---

**Algorithm 1:** The pseudocode to compute $\Delta(\mathcal{D})$ for a finite geometric dataset $\mathcal{D} = (X, \mu, F)$.

---

**Input** : Finite geometric dataset $\mathcal{D} = (X, \mu, F)$.
**Output:** $\Delta(\mathcal{D})$

**1 forall** $f$ *in* $F$ **do**
**2** $\quad\lfloor$ Compute feature sequence $l_{f,\mathcal{D}}$.
**3** $\Delta(\mathcal{D}) = 0$
**4 forall** $k$ *in* $\{2, \ldots, |X|\}$ **do**
**5** $\quad$ **forall** $f$ *in* $F$ **do**
**6** $\quad\quad\lfloor$ $\phi_{k,f}(\mathcal{D}) = \min_{j \in \{0, \ldots, |X|-k\}} l_{k+j}^{f,\mathcal{D}} - l_{1+j}^{f,\mathcal{D}}$.
**7** $\quad\lfloor$ $\Delta(\mathcal{D}) + = \max_{f \in F} \phi_{k,f}(\mathcal{D})$
**8** $\Delta(\mathcal{D}) = \frac{1}{|X|} \Delta(\mathcal{D})$
**9 return** $\Delta(\mathcal{D})$

---

solution for the computation of the ID. However, its quadratic runtime is obstructive for the application in large-scale data problems, which raises the necessity for a modification. We will present such a modification in the following section.

## 4 Intrinsic Dimension for Large-Scale Data

In order to speed up the computation of the ID we modify Algorithm 1 with regard to the accuracy of the result. Hence, we settle for an efficiently computable approximation of the ID. To give an overview over the necessary steps, we will

1. approximate the ID by replacing $\{2, \ldots, |X|\}$ in Line 4 of Algorithm 1 with a smaller subset $S \subseteq \{2, \ldots, |X|\}$, which we represent by $S := \{s_1, \ldots, s_l\}$. For all $k \notin S$, we will use $\{\phi_{s_1}(\mathcal{D}), \ldots, \phi_{s_l}(\mathcal{D})\}$ to estimate $\phi_k(\mathcal{D})$. This will eventually lead to two approximations of the ID, an underestimation and an overestimation.

2. compare the upper and lower approximation to provide an error bound of these approximations with respect to the exact ID. This error bound can be computed without knowing the exact ID.

3. argue how, the computation of the exact ID can be sped up with the help of knowledge about $\phi_{s_i}(\mathcal{D})$ for all $s_i \in S$. For this, we will in particular show that we can replace for all $k \in \{2, \ldots, |X|\} \setminus M$ the set $F$ with a subset $\hat{F}$, see Line 5-6 of Algorithm 1.

4. derive a formula which estimates the amount of computation cost which is saved by using only subsets of $F$ for the computation of the ID. This information can be used to estimate and decide whether the exact computation of the ID is computational feasible for a specific dataset.

The ensuing algorithm is shown in Algorithm 2. The underlying theory that justifies it is presented in the following. This theory will be based on the monotonicity of $i \mapsto \phi_{i,f}(\mathcal{D})$.

**Theorem 4.1.** *For $m > n \geq 2$ and $f \in F$ the following statements hold.*

1. *$\phi_{m,f}(\mathcal{D}) \geq \phi_{n,f}(\mathcal{D})$,*

2. *$\phi_m(\mathcal{D}) \geq \phi_n(\mathcal{D})$.*

*Proof.* The second inequality follows directly from the first one. Since per definition $\phi_{m,f}(\mathcal{D}) = \min_{M \subseteq X, |M|=m} \max_{x,y \in M} |f(x) - f(y)|$ and also $\phi_{n,f}(\mathcal{D}) = \min_{N \subseteq X, |N|=n} \max_{x,y \in N} |f(x) - f(y)|$, we need to show that for each $M \subseteq X$ with $|M| = m$ there exist $N \subseteq X$ with $|N| = n$ and $\max_{x,y \in M} |f(x) - f(y)| \geq \max_{x,y \in N} |f(x) - f(y)|$. It is sufficient to show that for $n = m - 1$. Choose $x_1, x_2 \in M$ such that

---

**Algorithm 2:** The pseudocode to compute $\Delta_{s,-}(\mathcal{D}), \Delta_{s,+}(\mathcal{D}), \Delta(\mathcal{D})$ for a finite GD $\mathcal{D} = (X, \mu, F)$.

---

**Input** : Finite GD $\mathcal{D} = (X, \mu, F)$, support sequence $s = (2 = s_1, \dots, s_l = |X|)$, *exact* (Boolean)
**Output:** $\Delta_{s,-}(\mathcal{D}), \Delta_{s,+}(\mathcal{D}), \Delta(\mathcal{D})$

**1 forall** $f$ *in* $F$ **do**
**2**     Compute feature sequence $l_{f,\mathcal{D}}$.

**3** $\Delta(\mathcal{D}) = 0$
**4** $s_0 = 1$
**5** $\phi_{s_0}(\mathcal{D}) = 0$
**6 forall** $i$ *in* $\{1, \dots, l\}$ **do**              // Iterate over support sequence indices
**7**     **forall** $f$ *in* $F$ **do**
**8**        $\phi_{s_i,f}(\mathcal{D}) = \min_{j \in \{0,\dots,|X|-s_i\}} l^{f,\mathcal{D}}_{s_i+j} - l^{f,\mathcal{D}}_{1+j}$      // Compute $\phi_{s_i,f}(\mathcal{D})$ with Lemma 3.5
**9**     $\phi_{s_i}(\mathcal{D}) = \max_{f \in F} \phi_{s_i,f}(\mathcal{D})$
**10**     $F_{s_{i-1}} = \{f \in F \mid \phi_{s_i,f}(\mathcal{D}) > \phi_{s_{i-1}}(\mathcal{D})\}$      // Compute $F_{s_{i-1}}$ for Lemma 4.7
**11**     $\Delta(\mathcal{D}) += \phi_{s_i}(\mathcal{D})$
**12** $\Delta_{s,-}(\mathcal{D}) = \frac{1}{|X|}(\Delta(\mathcal{D}) + \sum_{i=1}^{l-1}\sum_{s_i < j < s_{i+1}} \phi_{s_i}(\mathcal{D}))$     // Compute lower ID from Definition 4.2
**13** $\Delta_{s,+}(\mathcal{D}) = \frac{1}{|X|}(\Delta(\mathcal{D}) + \sum_{i=1}^{l-1}\sum_{s_i < j < s_{i+1}} \phi_{s_{i+1}}(\mathcal{D}))$     // Compute upper ID from Definition 4.2

    // Approximation finished. Continue with exact computation, if desired.
**14 if** *exact* **then**
**15**     **forall** $i$ *in* $\{1, \dots l\}$ **do**
**16**        **forall** $s_i < j < s_{i+1}$ **do**    // Iterate through all indices between two support elements
**17**           **forall** $f$ *in* $F_{s_i}$ **do**           // Only iterate through $F_{s_i}$ because of Lemma 4.7
**18**             $\phi_{s_i,f}(\mathcal{D}) = \min_{j \in \{0,\dots,|X|-s_i\}} l^{f,\mathcal{D}}_{s_i+j} - l^{f,\mathcal{D}}_{1+j}$.
**19**           $\phi_j(\mathcal{D}) = \max(\{\phi_{j,f}(\mathcal{D}) \mid f \in F_i\} \cup \{\phi_{s_i}\})$        // Use Lemma 4.7
**20**           $\Delta(\mathcal{D}) += \phi_j(\mathcal{D})$
**21**     $\Delta(\mathcal{D}) = \frac{1}{|X|}\Delta(\mathcal{D})$
**22**     **return** $\Delta_{s,-}(\mathcal{D}), \Delta_{s,+}(\mathcal{D}), \Delta(\mathcal{D})$
**23 return** $\Delta_{s,-}(\mathcal{D}), \Delta_{s,+}(\mathcal{D})$

---

$\max_{x,y \in M} |f(x) - f(y)| = |f(x_1) - f(x_2)|$. As $|M| > 2$ we find $x_3 \in M \setminus \{x_1, x_2\}$. Let $N := M \setminus \{x_3\}$. It holds that $\max_{x,y \in M} |f(x) - f(y)| = |f(x_1) - f(x_2)| = \max_{x,y \in N} |f(x) - f(y)|$. □

### 4.1 Computing Intrinsic Dimension via Support Sequences

Equipped with Theorem 4.1, we can bound $\Delta(\mathcal{D})$ and thus the intrinsic dimension through computing $\phi_{s_i}$ for a few $2 = s_1 < s_2 \cdots < s_l = |X|$.

**Definition 4.2.** *(Support Sequences and Upper / Lower ID) Let $s = (2 = s_1, \dots, s_{l-1}, s_l = |X|)$ be a strictly increasing and finite sequence of natural numbers. We call $s$ a support sequence of $\mathcal{D}$. We additionally define*

$$\Delta_{s,-}(\mathcal{D}) := \frac{1}{|X|}\left(\sum_{i=1}^{l} \phi_{s_i}(\mathcal{D}) + \sum_{i=1}^{l-1}\sum_{s_i < j < s_{i+1}} \phi_{s_i}(\mathcal{D})\right),$$

$$\Delta_{s,+}(\mathcal{D}) := \frac{1}{|X|}\left(\sum_{i=1}^{l} \phi_{s_i}(\mathcal{D}) + \sum_{i=1}^{l-1}\sum_{s_i < j < s_{i+1}} \phi_{s_{i+1}}(\mathcal{D})\right)$$

(5)

*and call accordingly* $\partial_{s,-}(\mathcal{D}) \coloneqq \frac{1}{\Delta_{s,+}(\mathcal{D})^2}$ *the* lower intrinsic dimension *of* $\mathcal{D}$ *and* $\partial_{s,+}(\mathcal{D}) \coloneqq \frac{1}{\Delta_{s,-}(\mathcal{D})^2}$ *the* upper intrinsic dimension *of* $D$.

The governing idea is for $i \in \{1,\ldots,l\}$ and $j$ with $s_i < j < s_{i+1}$ to substitute $\phi_j(\mathcal{D})$ with $\phi_{s_i}(\mathcal{D})$ or $\phi_{s_{i+1}}(\mathcal{D})$. With Theorem 4.1 this results in lower and upper bounds for $\Delta(\mathcal{D})$ and thus for the intrinsic ID. By comparing upper and lower bounds, we can approximate the ID and estimate the approximation error.

**Corollary 4.3.** *For support sequences $s$ holds $\Delta_{s,-}(\mathcal{D}) \leq \Delta(\mathcal{D}) \leq \Delta_{s,+}(\mathcal{D})$ and $\partial_{s,-}(\mathcal{D}) \leq \partial(\mathcal{D}) \leq \partial_{s,+}(\mathcal{D})$.*

**Definition 4.4** (Approximation Error)**.** *For a support sequence $s$ the* (relative) approximation error *of $\partial(\mathcal{D})$ with respect to $s$ is given by*

$$\mathrm{E}(s,\mathcal{D}) \coloneqq \frac{\partial_{s,+}(\mathcal{D}) - \partial_{s,-}(\mathcal{D})}{\partial_{s,-}(\mathcal{D})}.$$

With the computation of the upper and lower ID it is possible to bound the error with respect to the ID $\partial(\mathcal{D})$. The following corollary can be deduced from Corollary 4.3 and Definition 4.4.

**Corollary 4.5.** *For a support sequence $s$ the following statements hold.*

1. $\max\{\frac{\partial_{s,+}(\mathcal{D})-\partial(\mathcal{D})}{\partial(\mathcal{D})}, \frac{\partial(\mathcal{D})-\partial_{s,-}(\mathcal{D})}{\partial_{s,-}(\mathcal{D})}\}, \leq \mathrm{E}(s,\mathcal{D})$,

2. $\max\{|\partial_{s,+}(\mathcal{D}) - \partial(\mathcal{D})|, |\partial(\mathcal{D}) - \partial_{s,-}(\mathcal{D})|\} \leq |\partial_{s,+}(\mathcal{D}) - \partial_{s,-}(\mathcal{D})|$.

If the error of the approximation of a specific support sequence is not sufficient, further elements can be added to the support sequence. Directly from Equation (5) follows the following corollary.

**Corollary 4.6.** *Let $s = (2 = s_1,\ldots,s_l = |X|)$ be a support sequence and let $\hat{s} = (s_1,\ldots,s_i,p,s_{i+1},\ldots,s_l)$ with a support sequence with an additional element $p$. Then it holds that*

$$\Delta(\mathcal{D})_{\hat{s},-} = \Delta(\mathcal{D})_{s,-} + \frac{\sum_{p \leq j < s_{i+1}} \left((\phi_p(\mathcal{D})) - \phi_{s_i}(\mathcal{D})\right)}{|X|},$$

$$\Delta(\mathcal{D})_{\hat{s},+} = \Delta(\mathcal{D})_{s,+} + \frac{\sum_{s_i < j \leq p} \left((\phi_p(\mathcal{D})) - \phi_{s_{i+1}}(\mathcal{D})\right)}{|X|}.$$

For a given support sequence $s$, Corollary 4.5 gives us an upper bound for the error when $\partial_{s,+}(\mathcal{D})$ or $\partial_{s,-}(\mathcal{D})$ are used to approximate $\partial(\mathcal{D})$ without knowing $\partial(\mathcal{D})$. Hence, we can compute (a lower bound) for the accuracy when approximating the ID with Definition 4.4. As we can see in Section 5.1, Section 5.4 and Section 6, comparable small support sequences lead to sufficient approximations. Support sequences can also be used to shorten the computation of the exact intrinsic dimension as the following lemma shows.

**Lemma 4.7.** *Let $s = (2 = s_1,\ldots,s_l = |X|)$ be a support sequence. Furthermore, let $i \in \{1,\ldots,l-1\}$ and let $j \in \mathbb{N}$ with $s_i < j < s_{i+1}$. Let $F_{s_i} \coloneqq \{f \in F \mid \phi_{s_{i+1},f}(\mathcal{D}) > \phi_{s_i}(\mathcal{D})\}$. Then it holds that*

$$\phi_j(\mathcal{D}) = \max(\{\phi_{j,f}(\mathcal{D}) \mid f \in F_{s_i}\} \cup \{\phi_{s_i}(\mathcal{D})\}).$$

*Proof.* "$\geq$" follows from Theorem 4.1 and the definition of $\phi_j(\mathcal{D})$. "$\leq$" holds because for $f \in F \setminus F_{s_i}$ it holds that $\phi_{j,f}(\mathcal{D}) \leq \phi_{s_{i+1},f}(\mathcal{D})$, due to Theorem 4.1, and $\phi_{s_{i+1},f}(\mathcal{D}) \leq \phi_{s_i}(\mathcal{D})$, due to the construction of $F_{s_i}$. $\square$

Hence, given a specific $j$, it is possible to compute $\phi_j(\mathcal{D})$ using a subset of $F$. Based on the particular GD $\mathcal{D}$, this fact can considerably speed up the computation of the ID of $\mathcal{D}$, as we will see in Section 5.

An algorithm to approximate and compute the ID through support sequences is depicted in Algorithm 2. This algorithm takes as input a GD $\mathcal{D}$ and a chosen support sequence $s$. A reasonable choice for support sequences is discussed in Section 5.1. The output is $\Delta_{s,-}(\mathcal{D}), \Delta_{s,+}(\mathcal{D})$, and $\Delta(\mathcal{D})$, if desired (Line 14). In Line 2, all feature sequences are computed. In Line 6 to Line 11, $\phi_{s_i}(\mathcal{D})$ and $F_{s_{i-1}}$, as defined in Lemma 4.7, are computed. From Line 1 to Line 13, the feature sequences and the lower and upper ID are computed. If desired, the exact computation is done in Line 15 to Line 21. Here, we iterate for all support elements (Line 15) through all "gaps" between them (Line 16) and compute $\phi_j(\mathcal{D})$ using Lemma 4.7 (Line 17 to Line 19).

Table 1: Statistics of all datasets used in this work.

|  | Nodes | Edges | Attributes |
| --- | --- | --- | --- |
| PubMed | $19,717$ | $88,648$ | $500$ |
| Cora | $2,708$ | $10,556$ | $1,433$ |
| CiteSeer | $3,327$ | $9,104$ | $3,703$ |
| ogbn-arxiv | $169,343$ | $1,166,243$ | $128$ |
| ogbn-products | $2,449,029$ | $61,859,140$ | $100$ |
| ogbn-mag | $1,939,743$ | $21,111,007$ | $128$ |
| ogbn-papers100M | $111,059,956$ | $1,615,685,872$ | $128$ |

### 4.2 Estimating Computational Costs

Let $s = (s_1, \ldots, s_l)$ be a support sequence. After the computation of $F_{s_1}, \ldots, F_{s_{l-1}}$, we can estimate how much computation steps we can avoid in order to compute $\partial(\mathcal{D})$ with Algorithm 2 compared to Algorithm 1. Together with the error function $E(s)$, this estimation can help us to decide if it is desirable to compute the exact value $\partial(\mathcal{D})$ or leave it at $\partial_{s,-}(\mathcal{D})$ and $\partial_{s,+}(\mathcal{D})$. This is done in the following manner. For a specific $f \in F$, Lemma 3.5 shows that the computation of $\phi_{k,f}(\mathcal{D})$ requires $|X| - k + 1$ different subtractions and to keep the minimum value. Hence, the cost for computing $\Delta(\mathcal{D})$ and therefore $\partial(\mathcal{D})$ via Algorithm 1 can be estimated via $\mathcal{O}(|F| \sum_{k=2}^{|X|} (|X| - k + 1)) = \mathcal{O}(|F| \sum_{k=1}^{|X|-1} k) = \mathcal{O}(|F|(\frac{|X|^2 - |X|}{2}))$. However, if we use Algorithm 2, we solely have the cost to compute $\phi_{s_i}$. For all values $j$ with $s_i < j < s_{i+1}$, our cost estimation is $|F_{s_i}|(|X| - j + 1)$. Hence, for a given support sequence $s = (s_1, \ldots, s_l)$, we can estimate how many computations are saved using the following notions.

We address the *naive computation costs* for computing the ID of a GD with

$$\mathrm{C}(\mathcal{D}) := |F|(\frac{|X|^2 - |X|}{2}).$$

In contrast, for a support sequence $s = (s_1, \ldots, s_l)$ of $\mathcal{D}$, the *computation costs* are

$$\mathrm{C}_s(\mathcal{D}) := (|F| \sum_{k=1}^{l} |X| - s_k + 1) + \sum_{k=1}^{l-1} |F_{s_k}| \sum_{s_k < j < s_{k+1}} |X| - j + 1. \tag{6}$$

Hence, the *saved costs* of $s$ are

$$\mathrm{SC}_s(\mathcal{D}) := 1 - \frac{\mathrm{C}_s(\mathcal{D})}{\mathrm{C}(\mathcal{D})}.$$

Once we have computed $\phi_{s_i}(\mathcal{D})$ and $F_i$, depending on the saved costs, we can decide to discard the support sequence or to continue further computations with it. Furthermore, using the error estimation, we can decide to compute the exact ID or to settle with the approximation.

## 5 Intrinsic Dimension of Graph Data

Graph data is of major interest in the realm of geometric learning and beyond. In the following, a *graph dataset* $D = (\mathcal{X}, G)$ consists of an undirected, unweighted graph $G = (V, E)$, where $V = \{v_1, \ldots, v_n\}$ is a finite set of vertices, $E \subseteq \binom{V}{2}$ and $\mathcal{X} \in \mathbb{R}^{n \times d}$ is a *d-dimensional attribute matrix*. The row-vector $\mathcal{X}_i = (\mathcal{X}_{i,1}, \ldots, \mathcal{X}_{i,d})$ is called the *attribute vector* of $v_i$.

Learning from such data is often done via *graph neural networks*. The idea is to extend common multi-layer perceptrons by a so called *neighborhood aggregation*, where internal representations of graph neighbors are combined at specific layers. In earlier works, neighborhood aggregation is done at multiple layers (Kipf & Welling, 2017; Hamilton et al., 2017; Velickovic et al., 2018). Due to scalability, recent approaches perform multiple iterations of neighborhood aggregation as a preprocessing step and then use the aggregated features

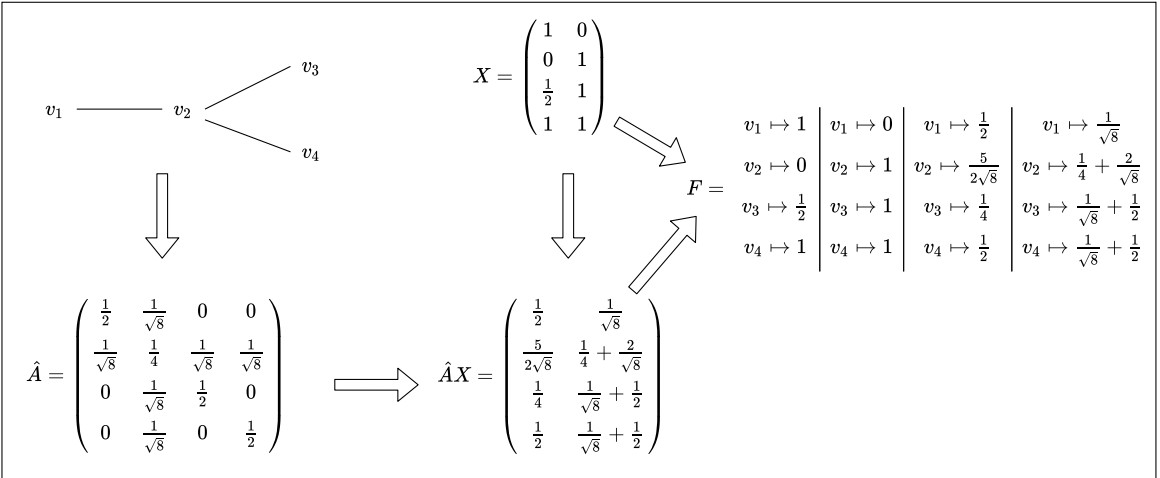

Figure 1: Example of an $k$-hop geometric dataset with $k = 1$. Given are a graph and an attribute matrix $X$. Then, the normalized adjacency matrix $\hat{A}$ and then $\hat{A}X$ are computed. The feature set $F$ consists of the coordinate projections of $X$ and $\hat{A}X$. In the figure, every column after "$F =$" represents one $f \in F$. Note, that in this example the normalization factor $\frac{1}{d_{\max}}$ is 1.

as combined input (Rossi et al., 2020; Sun & Wu, 2021; Zhang et al., 2021). To be more specific, the employed networks use inputs of the form

$$(\mathcal{X}, \hat{A}\mathcal{X}, \hat{A}^2\mathcal{X} \ldots, \hat{A}^k X), \tag{7}$$

where $\hat{A}$ is a *transition matrix* that is derived from the graph structure. The most common choice for such a matrix is the normalized adjacency matrix, i.e., $\hat{A}_{i,j} = (\sqrt{\deg(v_j)\deg(v_i)})^{-1}$ if $v_j \in N(v_i)$ and $\hat{A}_{i,j} = 0$ else. Here, $N(v_i) \coloneqq \{v_j \in V \mid \{v_i, v_j\} \in E\} \cup \{v_i\}$ is the set of neighbors of $v_i$ and $\deg(v_i) \coloneqq |N(v_i)|$ is the node degree of $v_i$. The feature set of the following geometric dataset corresponds to the input in Equation (7).

**Definition 5.1.** *Let $k \in \mathbb{N}$ and $\hat{A}$ be the normalized adjacency matrix of a graph dataset $D$. Furthermore, let $d_{\max} \coloneqq \max_{j \in \{1,\ldots,d\}} \max_{i,k \in \{1,\ldots,n\}} |\mathcal{X}_{i,j} - \mathcal{X}_{k,j}|$. We call the set*

$$F_{D,k} \coloneqq \{v_i \mapsto \frac{1}{d_{\max}}(\hat{A}^m \mathcal{X})_{i,j} \mid m \in \{0,\ldots,k\}, j \in \{1,\ldots,d\}\}$$

*the $k$-hop feature functions of $D$. Let $\nu$ be the normalized counting measure on $V$. If there exist for each $v_i, v_k \in V$ with $v_i \neq v_j$ elements $m \in \{0,\ldots,k\}, j \in \{1,\ldots,d\}$ such that $\frac{1}{d_{\max}}(\hat{A}^m \mathcal{X})_{i,j} \neq \frac{1}{d_{\max}}(\hat{A}^m \mathcal{X})_{k,j}$, then $\mathcal{D}_k = (V, F_{D,k}, \nu)$ is a GD. We call it the $k$-hop geometric dataset of $D$.*

Basic statistics of all seven graph datasets considered in the following sections are depicted in Table 1. The statistics for **Cora**, **PubMed** and **CiteSeer** were taken from PyTorch Geometric [2]. The statistics of the OGB datasets were taken from the Open Graph Benchmark. [3] An example of a $k$-hop geometric dataset is depicted in Figure 1. It is well-known that the normalized adjacency matrix $\hat{A} \in \mathbb{R}^{n \times n}$ of a graph has a spectral radius of 1. As $\hat{A}$ is symmetric, this yields $\|\hat{A}x - \hat{A}y\| \leq \|x - y\|$ for $x, y \in \mathbb{R}^n$. The significance of this property is for the respective computations, however, limited, since it primarily leads to insights of the behavior of the columns of $\mathcal{X}$ under multiplication with powers of $\hat{A}$. In contrast, the attribute vectors of the vertices are represented via the rows. Moreover, we may point out that we are not considering the Euclidean distances between attribute vectors, but differences between coordinate values. Thus, the spectral radius of $\hat{A}$ does not provide direct insights into $F_{D,k}$.

---

[2] https://pytorch-geometric.readthedocs.io/en/latest/modules/datasets.html#torch_geometric.datasets. Planetoid

[3] https://ogb.stanford.edu/docs/nodeprop

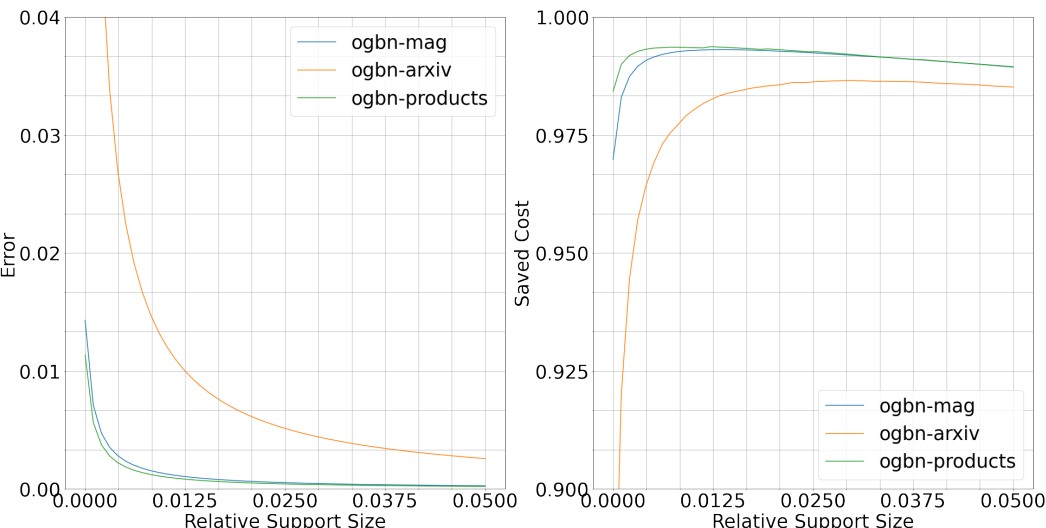

Figure 2: Errors and saved costs for approximating and computing the intrinsic dimensionality for $2 - hop$ geometric datasets with different lengths of the support sequence.

## 5.1 Choosing Support Sequences

Algorithm 2 relies on a proper choice for a support sequence $s$. To choose $s$, two properties have to be considered. Namely, the length of the support sequence and the spacing of the elements. Regarding the second point, we decided to use log-scale spacing. To get such a support sequence for a geometric dataset $\mathcal{D} = (X, F, \mu)$, we first choose a geometric sequence $\hat{s} = (s_1, \ldots s_l)$ of length $l$ from $|X|$ to 2. We derive the final support sequence $s$ from $s' = (\lfloor |X| + 2 - s_1 \rfloor, \ldots, \lfloor |X| + 2 - s_l \rfloor)$ by removing duplicated elements.

In the following, we study the error and the saved costs for different lengths $l$ of the support sequence. Here, for a geometric dataset, we investigate how $\text{E}(s, \mathcal{D})$ and $\text{SC}(s)$ vary for $s$ chosen with $l \in \{\lfloor 0.001 * |X| \rfloor, \lfloor 0.002 * |X| \rfloor, \ldots, \lfloor 0.05 * |X| \rfloor\}$. Here, if $l = \lfloor r * |X| \rfloor$, we call $r \in \mathbb{R}$ the *relative support size* of the resulting support sequence $s$. We experiment with common benchmark datasets, namely **ogbn-arxiv**, **ogbn-mag** and **ogbn-products** from the Open Graph Benchmark (Hu et al., 2020; 2021). Since for **ogbn-mag** only a subset of vertices is equipped with attribute vectors, we generate the missing vectors via metapath2vec (Dong et al., 2017). For all datasets, we consider the 2-hop geometric dataset. The results are depicted in Figure 2.

### 5.1.1 Results

For all datasets, low errors and high saved costs can be reached with a remarkably short support sequence. With relative support sizes of under 0.015 all datasets are approximated with an accuracy of over 99%. Furthermore, the saved costs for sequences with comparable relative support sizes is over 0.98. It stands out, that for the larger datasets **ogbn-mag** and **ogbn-products**, shorter sequences (relative to the size of the dataset) lead to lower errors and higher saved costs then for **ogbn-arxiv**. Our results further indicate, that a relative support size between 0.01 and 0.02 is a reasonable range for maximizing the saved costs. For longer support sequences, the saved cost decrease while the error does not change dramatically, at least for the 2-hop geometric datasets of **ogbn-mag** and **ogbn-products**. Note, that longer support sequence do not always lead to a higher amount of saved costs. For longer support sequences $s$ the costs of computing $\phi_k(\mathcal{D})$ for $k \notin s$ decreases. However, the costs of computing $\phi_{s_i}(\mathcal{D})$ for all elements $s_i \in s$ increase.

Table 2: Intrinsic dimension and performances on classification tasks. In the upper table, we display IDs for all $k$-hop geometric datasets for $k \in \{0, \ldots, 5\}$. In the middle table, we display the ID estimated by the MLE baseline. In the lower table we display mean and standard derivations for test accuracy of a standard SIGN model on the classification tasks which belongs to the dataset.

| $k$-hop / Dataset | 0 | 1 | 2 | 3 | 4 | 5 |
|---|---|---|---|---|---|---|
| PubMed | 2542.3425 | 2336.6611 | 2077.5821 | 2077.0953 | 2077.0886 | 2077.0848 |
| Cora | 6.2523 | 3.8324 | 3.6689 | 3.6627 | 3.6624 | 3.6623 |
| CiteSeer | 22.3337 | 11.3166 | 10.2347 | 9.8134 | 9.5491 | 9.3795 |
| ogbn-arxiv | 83.9160 | 31.4731 | 31.4731 | 31.4730 | 30.7370 | 30.3767 |
| ogbn-products | $1,169,323.2496$ | $1,169,044.4736$ | $1,169,044.2216$ | $1,169,044.2216$ | $1,169,044.2216$ | $1,169,044.2216$ |
| ogbn-mag | $2,311.3509$ | $2,284.0290$ | $2,284.0290$ | $2,284.0290$ | $2,284.0290$ | $2,284.0290$ |

| $k$-hop / Dataset | 0 | 1 | 2 | 3 | 4 | 5 |
|---|---|---|---|---|---|---|
| PubMed | 24.4623 | 24.7303 | 23.3924 | 22.2779 | 21.3495 | 20.5642 |
| Cora | 30.6049 | 28.1785 | 19.9316 | 10.8186 | 9.2970 | 8.6155 |
| CiteSeer | 58.9593 | 26.5031 | 16.5556 | 12.0495 | 9.3171 | 7.9572 |
| ogbn-arxiv | 16.2948 | 19.8571 | 18.9068 | 18.2265 | 17.4905 | 16.9325 |
| ogbn-products | 2.8694 | 4.7542 | 4.7950 | 4.7659 | 4.6943 | 4.6687 |
| ogbn-mag | 30.7024 | 33.2848 | 31.5140 | 30.4844 | 29.9080 | 29.5956 |

| $k$-hop / Dataset | 0 | 1 | 2 | 3 | 4 | 5 |
|---|---|---|---|---|---|---|
| PubMed | $.6850 \pm .0145$ | $.7191 \pm .0123$ | $.7378 \pm .0362$ | $.7565 \pm .0165$ | $.7615 \pm .0160$ | $.7571 \pm .0234$ |
| Cora | $.5329 \pm .0120$ | $.7223 \pm .0117$ | $.7766 \pm .0045$ | $.7870 \pm .0076$ | $.7917 \pm .0084$ | $.7951 \pm .0047$ |
| CiteSeer | $.4975 \pm .0075$ | $.6165 \pm .0160$ | $.6530 \pm .0101$ | $.6677 \pm .0074$ | $.6695 \pm .0085$ | $.6734 \pm .0080$ |
| ogbn-arxiv | $.5341 \pm .0090$ | $.6572 \pm .0052$ | $.6903 \pm .0056$ | $.6917 \pm .0074$ | $.6901 \pm .0083$ | $.6890 \pm .0051$ |
| ogbn-products | $.5969 \pm .0016$ | $.7204 \pm .0017$ | $.7590 \pm .0017$ | $.7660 \pm .0014$ | $.7678 \pm .0022$ | $.7687 \pm .0019$ |
| ogbn-mag | $.2712 \pm .0020$ | $.3635 \pm .0029$ | $.3879 \pm .0030$ | $.3959 \pm .0029$ | $.3983 \pm .0050$ | $.4012 \pm .0040$ |

## 5.2 Neighborhood Aggregation and Intrinsic Dimension

We study how the choice of $k$ affects the intrinsic dimension value of the $k$-hop geometric dataset. For this, we compute the intrinsic dimension for $k \in \{0, 1, \ldots, 5\}$ for six datasets: the three datasets mentioned above and **PubMed**, **Cora** and **CiteSeer** (Yang et al., 2016), which we retrieved from PyTorch Geometric (Fey & Lenssen, 2019). Furthermore, we train GNNs which use the feature functions of $k$-hop geometric datasets as information for training and inference. This allows us to discover connections between the ID of specific datasets with respect to the considered feature functions and the performance of classifiers, which rely on these feature functions. For this, we train SIGN models (Rossi et al., 2020) for $k \in \{0, \ldots, 5\}$. Implementation details and parameter choices can be found in Appendix A.1.

### 5.2.1 Baseline Estimator

To investigate to which extent our ID function surpasses established ID estimators with respect to estimating the discriminability of a dataset, we also compute all ID values with the Maximum Likelihood Estimator (MLE) (Levina & Bickel, 2004). This estimator is commonly used in the realm of deep learning (Pope et al., 2020; Ma et al., 2018a;b). For our experiments, we use the corrected version proposed by MacKay & Ghahramani (2005). Note, that the MLE is only applicable to datasets $\mathcal{X} \in \mathbb{R}^{n \times d}$ and is thus not able to respect the neighborhood aggregated feature functions. Hence, we incorporate the neighborhood information of a $k$-hop dataset by concatenating feature vectors with the neighborhood aggregated feature vectors. Due to performance reasons, only subsets of the data points are considered for **ogbn-mag** and **ogbn-products**. More details to our usage of the MLE are discussed in Appendix A.2.

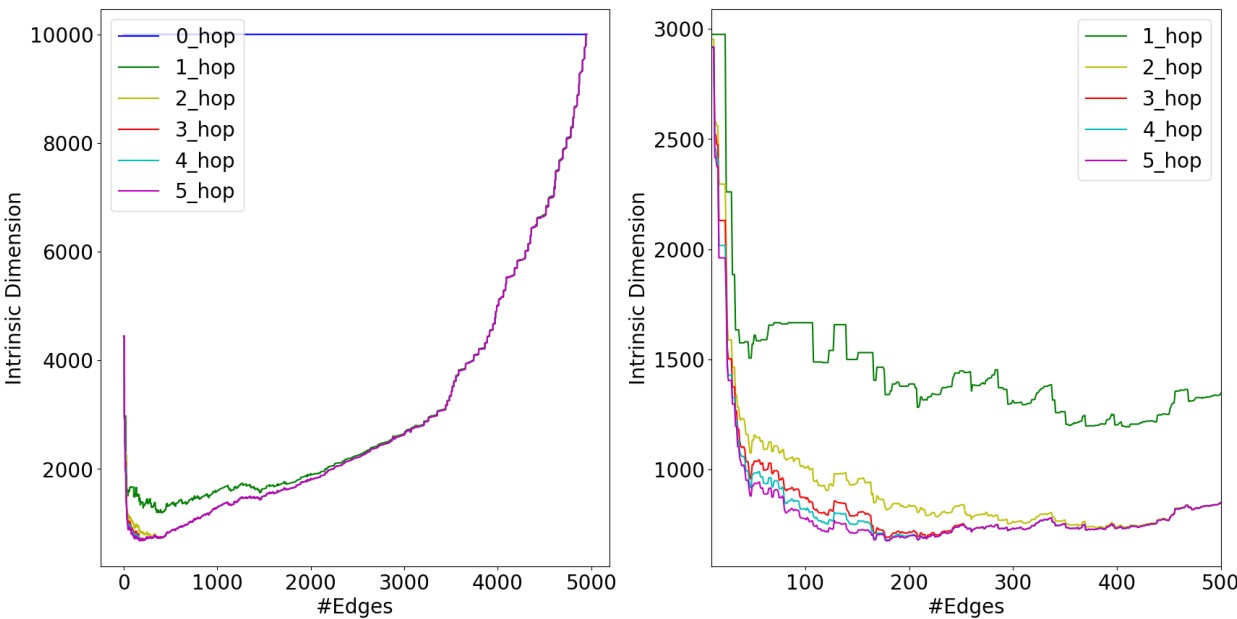

Figure 3: Intrinsic dimension of one-hot encoded features. The right plot is a zoom of the left one which hides the $0-$hop geometric dataset.

### 5.2.2 Results

We find that one iteration of neighborhood aggregation always leads to a huge drop of the ID when using our ID function. However, consecutive iterations only lead to a small decrease. For the datasets from OGB, some iterations lead to no drop of the ID dimension at all. For **ogbn-mag**, only the first iteration significantly decreases the ID, for **ogbn-products**, only the first two iterations are relevant for decreasing the ID. It stands out, that for **ogbn-arxiv**, the second and third iteration lead to no significant decrease, but the fourth and fifth do. The results for **PubMed** stand out. Here, the second iteration of neighborhood aggregation leads to a comparable decrease as the first one.

Considering the classification performances, the first iteration is again the key factor, leading to a significant increase in accuracy. As for the ID, the **PubMed** dataset behaves differently than the other datasets: the second iteration of neighborhood aggregation leads to a comparable increase in accuracy as the first.

The MLE ID behaves different. Here, no pattern of the first iteration of aggregation being the key for decreasing the data complexity is observed. For some datasets, the first rounds of feature aggregation may even increase the intrinsic dimension. To sum up, our results indicate that our ID is a better indicator for classification performance then the MLE ID.

### 5.3 Synthetic Data

To get further insights into the behavior of our ID notion, we now consider $k-$hop geometric dataset for one-hot encoded graph data, i.e., $\mathcal{D} = (\mathcal{X}, (V, E))$, with $\mathcal{X} = \mathbb{I}_{|V|}$ where $\mathbb{I}_{|V|}$ is the $|V|$-dimensional identity matrix. We consider the case of $V = \{1, \dots, 100\}$ and determine the ID for $k \in \{0, \dots, 5\}$ for increasing edge sizes. To do so, we place the 4950 possible edges in a random order and add them step by step and compute the ID notions for the $k-$hop geometric dataset. The results can be found in Figure 3.

### 5.3.1 Results

The $0-$hop geometric dataset has an ID which does not depend on the amount of edges. This is not surprising since it does not incorporate any graph information. For all other $k$ values, the ID first sharply decreases and then increases. This indicates that the addition of neighborhood aggregation is particularly useful for

Table 3: Approximation of intrinsic dimension for ogbn-papers100M.

| $k$ | 0 | 1 | 2 | 3 | 4 | 5 |
|---|---|---|---|---|---|---|
| $\partial_{s,-}(\mathcal{D})$ | 282.2380 | 171.7385 | 148.3323 | 137.7662 | 128.2751 | 125.3418 |
| $\partial_{s,+}(\mathcal{D})$ | 282.3387 | 171.7997 | 148.3852 | 137.8153 | 128.3208 | 125.3864 |
| $E(s,\mathcal{D})$ | 0.0004 | 0.0004 | 0.0004 | 0.0004 | 0.0004 | 0.0004 |

Table 4: Error on randomly generated data.

| $n$ | $d$ | $E(s,\mathcal{D})$ |
|---|---|---|
| $10^6$ | 10 | $2.55 * 10^{-4} \pm 1.13 * 10^{-8}$ |
| $10^6$ | 50 | $2.55 * 10^{-4} \pm 5.36 * 10^{-9}$ |
| $10^6$ | 250 | $2.55 * 10^{-4} \pm 2.31 * 10^{-9}$ |
| $10^7$ | 10 | $3.08 * 10^{-4} \pm 4.78 * 10^{-10}$ |
| $10^7$ | 50 | $3.08 * 10^{-4} \pm 6.57 * 10^{-10}$ |
| $10^7$ | 250 | $3.08 * 10^{-4} \pm 6.16 * 10^{-10}$ |
| $10^8$ | 10 | $3.55 * 10^{-4} \pm 3.67 * 10^{-11}$ |
| $10^8$ | 50 | $3.55 * 10^{-4} \pm 1.34 * 10^{-11}$ |
| $10^8$ | 250 | $3.55 * 10^{-4} \pm 2.69 * 10^{-11}$ |

graphs of moderate density. Here, the addition of additional rounds of aggregation beyond the first one can further lower the ID. For higher edge sizes, the ID difference between different $k$ values vanishes.

### 5.4 Approximation of Intrinsic Dimension on Large-Scale Data

To demonstrate the feasibility of our approach, we use it to approximate the ID of the well known, large-scale **ogbn-mag-papers100M** dataset. For this, we construct the support sequence as in Section 5.1 with $l = 100.000$. The results are depicted in Table 3. On our *Xeon Gold System* with 16 cores, approximating the ID of a $k$-hop geometric dataset build from **ogbn-mag-papers100M** is possible within a few hours. While the ID drops for every iteration of neighborhood aggregation, the decrease becomes smaller. The ID of the different $k$-hops can be differentiated by the approximation, i.e., $\partial_{s,-}(\mathcal{D}_i) > \partial_{s,+}(\mathcal{D}_{i+1})$ for $i \in \{0,\dots,4\}$. It stands out, that even for such a short support sequence (compared to the size of the dataset), the observed error is remarkably low. In detail, we can approximate the ID with an accuracy of over 99.95%. It is further remarkable, that the error does not change significantly for different $k$. We observed this effect also for the other datasets. Our results on **ogbn-papers100M** indicate, that with short support sequences, we can sufficiently approximate the ID of large-scale graph data.

## 6 Errors of Random Data

To further understand how our approximation procedure behaves we conducted experiments on random data. We considered different data sizes and different amount of attributes. For this, we experimented with real-valued datasets, i.e. datasets represented by an attribute matrix $\mathcal{X} \in \mathbb{R}^{n \times d}$. Here, the feature functions are given by the data columns. To be more detailed, the considered geometric dataset is $\mathcal{D} = (\{\mathcal{X}_i \mid i \in \{1,\dots,n\}\}, \{X_i \mapsto X_{i,j} \mid j \in \{1,\dots,d\}\}, \nu)$. Here, $\nu$ is again the normalized counting measure and $\mathcal{X}_i$ is he $i-th$ row vector of $\mathcal{X}$. We iterate $n$ through $\{10^6, 10^7, 10^8\}$ and $d$ through $\{10, 50, 250\}$. We repeat all experiments 3 times. For all datasets, we build a support sequence as described in Section 5.1 with $l = 100,000$. The results can be found in Table 4.

For all datasets, the errors are small and the accuracy is over 99.9% for all considered data sizes. The difference in the error for different values of $d$ is negligible. Furthermore, we have small standard deviations. All this indicates that $l = 100,000$ is a reasonable default choice that leads to sufficient approximations in a large range of data and attribute sizes.

# 7 Conclusion and Future Work

We presented a principle way to efficiently compute the intrinsic dimension (ID) of geometric datasets. Our approach is based on an axiomatic foundation and accounts for underlying structures and is therefore especially tailored to the field of geometric learning. We proposed a novel speed up technique for an algorithm which has quadratic complexity with respect to the amount of data points. This enabled us to compute the ID of several real-world graphs with up to millions of nodes. Equipped with this ability, we shed light on connections of classification performances of graph neural networks and the observed intrinsic dimension for common benchmark datasets. Finally, using a novel approximation technique, we were able to show that our method scales to graphs with over 100 million nodes and billions of edges. We illustrated this by using the well-known **ogbn-papers100M** dataset.

Future work includes the identification of suitable feature functions for other domains, such as learning on text or image data. Incorporating the structure of such datasets into the computation of intrinsic dimensionality is an open research problem. Another promising research direction is to investigate how the ID of datasets could be manipulated. Since our investigations suggest connections between a low ID and high classification performances, this has the potential to enhance learning procedures.

### Acknowledgement

The authors thank the State of Hesse, Germany for funding this work as part of the LOEWE Exploration project "Dimension Curse Detector" under grant LOEWE/5/A002/519/06.00.003(0007)/E19.

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

# A   Appendix

## A.1   Setup of SIGN classifiers

For **PubMed**, **Cora** and **CiteSeer**, we train on the classification task provided by Pytorch Geometric (Fey & Lenssen, 2019) which was earlier studied by Yang et al. (2016). All Open Graph Benchmark datasets are trained and tested on the official *node property prediction* task.[4] Our goal is not to find optimal classifiers but to discover connections between the choice of $k$, the ID and classifier performance. Thus, we omit excessive parameter tuning and stick to reasonable standard parameters. For all tasks, we use a simple SIGN model Rossi et al. (2020) with one hidden inception layer and one classification layer. For **PubMed**, **CiteSeer** and **Cora**, we use batch sizes of 256, hidden layer size of 64 and dropout at the input and hidden layer with 0.5. The learning rate is set to 0.01. All these parameters were taken from Kipf & Welling (2017). For **ogbn-arxiv**, **ogbn-mag** and **ogbn-products**, we stick to the parameters from the SIGN implementations on the OGB leaderbord. For **ogbn-arxiv**, we use a hidden dimension of 512, dropout at the input with 0.1 and with 0.5 at the hidden layer. For **ogbn-mag**, we use a hidden dimension of 512, do not dropout at the input and use dropout with 0.5 at the hidden layer. For **ogbn-products**, we use a hidden dimension of 512, input dropout of 0.3 and hidden layer dropout of 0.4. For all ogbn tasks, the learning rate is 0.001 and the batch-size 50000. For all experiments, we train for a maximum of 1000 epochs with early stopping on the validation accuracy. Here, we use a patience of 15. These are the standard parameters of Pytorch Lightning.[5] For all models, we use an Adam optimizer with weight decay of 0.0001. We report mean test accuracies over 10 runs. The intrinsic dimensions and the test accuracy are shown in Table 2.

## A.2   Details on Baseline ID Estimator

To use the MLE ID, we have to convert the $k$-hop geometric dataset $(V, F_{D,k}, \nu)$ of graph data $D = (\mathcal{X}, G)$, where $\mathcal{X} \in \mathbb{R}^{n \times d}$ into a real-valued feature matrix $\hat{X}$. This done by concatenating the rows of $\mathcal{X}$ with the rows of $\hat{A}\mathcal{X}, \ldots \hat{A}^k \mathcal{X}$, i.e., $\hat{X} \in \mathbb{R}^{n \times (k+1)d}$ with

$$\hat{X}_{i,j} \coloneqq \begin{cases} \mathcal{X}_{i,j} & j \in \{1, \ldots, d\}, \\ (A^n \mathcal{X})_{i,\hat{j}} & j = nd + \hat{j} \ \texttt{for} \ n \in \{1, \ldots, k\}, \hat{j} \in \{1, \ldots d\}. \end{cases}$$

The MLE is given via

$$\text{MLE}(\hat{X}) \coloneqq \frac{1}{n(k-1)} \sum_{i=1}^{n} \sum_{j=1}^{l-1} \log\left(\frac{d(\hat{X}_i, N_l(\hat{X}_i))}{d(\hat{X}_i, N_j(\hat{X}_i))}\right), \tag{8}$$

where $d$ is the euclidean metric and $N_j(\hat{X}_i)$ is the $j$-th nearest neighbor of $\hat{X}_i$ with respect to the Euclidean metric. Thus, the MLE depends on a parameter $l$, which we set to 5.

We implement the MLE by using the *NearestNeighbors* class of scikit-learn (Pedregosa et al., 2011) and then building the mean of all $\log\left(\frac{d(\hat{X}_i, N_5(\hat{X}_i))}{d(\hat{X}_i, N_j(\hat{X}_i))}\right)$ with $i \in \{1, \ldots, n\}$ and $j \in \{1, \ldots, 5\}$. Here, we skip all elements where $d(\hat{X}_i, N_j(\hat{X}_i)) = 0$. This can happen, when $\hat{X}$ has duplicated rows, representing data points with equal attribute vectors.

For **ogbn-mag** and **ogbn-products**, computing Equation (8) is not possible due to performance reasons. Here, we sample $169,343$ indices[6] $I \subset \{1, \ldots, n\}$ and only compute

$$\text{MLE}(\hat{X}) \coloneqq \frac{1}{n(k-1)} \sum_{i \in I} \sum_{j=1}^{l-1} \log\left(\frac{d(\hat{X}_i, N_l(\hat{X}_i))}{d(\hat{X}_i, N_j(\hat{X}_i))}\right).$$

.

---

[4]`https://ogb.stanford.edu/docs/nodeprop/`
[5]`https://www.pytorchlightning.ai/`
[6]This is the amount of nodes of ogbn-arxiv, the largest network for which the full computation was feasible.

