# OpenReview forum: "Intrinsic Dimension for Large-Scale Geometric Learning"
_TMLR — Accepted by TMLR_

### Review · Reviewer_pQDm · 2022-10-28

**Summary Of Contributions:**

In this work, the authors study a notion of intrinsic dimension (ID) for finite datasets, based on how discriminative some given features are. This is an instanciation of a notion of ID proposed is earlier work related to some notion of concentration, to the particular case of finite sets and counting measure.

The authors first propose a polynomial algorithm to compute the ID, based on some reformulation that uses sorted sequences of features. Since the resulting algorithm is still quadratic, they propose a faster version with subsampling, as well as a way to bound the resulting error and evaluate savings of computation cost.

They then apply their methodology to graph data, where the features are taken as a concatenation of node features smoothed by the normalized adjacency matrix at several order. They find that, for different order, the ID is somewhat correlated with the performance of a GNN classifier.

**Audience:**

Yes

**Claims And Evidence:**

Yes

**Requested Changes:**

I think synthetic experiments and better motivation could help improving the paper, but they are not critical to my acceptance.

Some minor comments/typos:
- p3, top: the choice of a measure $\mu$ on $\mathbb{R}$ is confusing, it was a measure on $X$ two paragraphs earlier. Also it is confused with $\nu$ after
- the notation $-\alpha$ (instead of just $\alpha$) is weird, is it motivated in earlier work?
- the "without loss of generality" in the proof of Lemma 3.1 is not so trivial, maybe it is possible to be more rigorous here
- unless I am mistaken, in Lemma 3.4, $\phi$ is the minimum of the set, not simply a member of. It is computed that way in the algorithm.
- page 5 bottom: "$k \notin$", missing $S$
- $s$ should be $S$ in algorithm 2
- the notation $\sum_{s_i<l<s_{i+1}} \phi_{s_i}$ is confusing, isn't it just $(s_{i+1}-s_i - 1)$ times $\phi_{s_i}$ ?
- I think theorem 4.1 is immediate, by definition of $\phi$ as a minimum over $M$ of size $m$ or $n$, no need for a proof.
- It is quite counter-intuitive that the saved computation cost is not strictly decreasing with the size of the support sequence, it needs to be written and emphasized
- Maybe the table 2 of results could be better presented as curves, that will better outline the potential correlation between ID and performance ?


**Strengths And Weaknesses:**

Strengths:
- the paper is quite complete, and the experiments pretty thorough
- the study of different notions of ID is particularly relevant in the current landscape, especially for geometric data where classical ones such as VC-dimension or Rademacher complexities may be difficult to define
- the paper is clear and well-written

Weaknesses:
- the proposed ID could be somewhat better motivated. At the end of the day, it is difficult to interpret and to get an intuition behind the performed computations, unlike some other classical notions of dimension such as the intrinsic dimension of a supporting manifold or the relationship between separability and VC dimension. The intuition behind the "concentration properties" and "axioms" are somewhat quickly skipped, they could be better detailed, even if appearing in earlier work.
- while quite thorough, the experiments on real data are not very convincing as to the definitive existence of a link between the proposed ID and learning performance; a singular drop (or increase) in ID (or performance) between k=0 and k=1 is far from being surprising for graph data. The difference between the behavior of ID and MLE is not so striking, especially given that the ID has extremely different range depending on the dataset (eg a drop from 1169323 to 1169044 is small in relative term). Maybe some experiments on synthetic data could help both better motivate the proposed ID and give better intuition ?
- the choice of support sequence is quite arbitrary. Can computation be done when $S$ is a stream, where the user adds new $s_l$ sequentially until a desired accuracy is reached ?

---

> ### Author Response · Authors · 2022-11-15
> **Response to Review**
>
> Dear Reviwer pGDm,
> we thank you for your detailed review. It really helped us to enhance our paper.
> Regarding your remarks and questions:
>
> >the proposed ID could be somewhat better motivated.[...]The intuition behind the "concentration properties" and "axioms" are somewhat quickly skipped, they could be better detailed, even if appearing in earlier work.
>
> Pestov defines the curse of dimensionality as "[...] a name given to the situation
> where all or some of the important features of a dataset
> sharply concentrate near their median (or mean) values and thus
> become non-discriminating. In such cases, X is perceived as
> intrinsically high-dimensional." [1] Thus, the ID estimator aims to compute to which extent the features allow to discriminate different data points. For a specific feature f \in F, [2] therefore defines the Partial Diameter of $f$ with regard to a specific $\alpha$ such that it displays to which extent $f$ can discriminate subsets with minimal measure $1-\alpha$. The observable diameter with respect to $\alpha$ then defines to which extent $F$ can discriminate points with minimum measure 1-$\alpha$ by being defined as the supremum of the partial diameter of all $f \in F$. To observe the discriminability for different minimal measures $\alpha$, we define $\Delta(\mathcal{D})$ by integrating over $\alpha$ in the range from zero to one. Thus, higher values of $\Delta\mathcal{D}$ indicate a high discriminability and hence a lower intrinsic dimensionality with respect to Pestovs definition above. Furthermore, the ID $\partial (\mathcal{D}) = \frac{1}{\Delta(\mathcal{D})}²$ respects the formal axiomatization [2] for ID functions, which we added to the paper.
>
> To sum up, the proposed ID measure does NOT approximate the dimensionality of a manifold in the Euclidean sense, but directly reflects to which extent a geometric data set is effected by the curse of dimensionality.
>
> This extended explanation and a high level description of the different axioms has been added to the paper.
>
> >while quite thorough, the experiments on real data are not very convincing [..]. Maybe some experiments on synthetic data could help both better motivate the proposed ID and give better intuition ?
>
>  We agree, that the drop from k=0 to k=1 is not surprising and should be observed coming from the definition of intrinsic dimensionality by Pestov. However, our experiments show that MLE does not capture this drop. We will further add the following two parts to the revised version to get further intuition about how our id measure behaves.
> - On the theoretical side, we will explain the link between the ID and the process of adding features. To be more specific we show that adding features that are incapable of further discriminating data does not decrease the ID. In particular, constant feature functions and features that are permutations of already existing feature functions do not decrease the ID.
> - On the empirical side, we add further experiments on synthetic data. To be more specific,
> we add experiments on one-hot encoded data consisting of node size 100. We start with an empty edge set and add edges in a random order until we have the complete graph. In every step, we determine the ID of the k-hop geometric dataset at every step with k from 0 to 5.
>
> We hope that both this parts help to get a better understanding of our method.
>
> > the choice of support sequence is quite arbitrary. Can computation be done when is a stream, where the user adds new sequentially until a desired accuracy is reached ?
>
>  Note, that the choice of the support sequence is secondary if we do not stop with the approximations but continue with the exact computation, as done with all datasets in the paper except ogbn-papers100M. In these cases, the result does NOT depend on the specific choice of the support sequence. Furthermore, we conducted additionall experiments on random data to get further empirical evidence, that a support sequence with $l=100,000$ is a general reasonable default choice:
>
>  We experimented with 1 milltion, 10 million and 100 million data points with 10, 50 or 250 attributes. In all cases, we have an accuracy of over 99,9% with a geometric support sequence with l=100,000.
>
>   We thank the reviewer for pointing us towards stream data. We have not considered those in our initial version. However, we indeed can handle stream data: In Equation 5 in Definition 4.2, only a few summands are changed when another element is added into a support sequence.
>
>  We added the random experiments mentioned above and the arguments regarding streams to the revised version.
>
> [1] Pestov, Vladimir. "Intrinsic dimension of a dataset: what properties does one expect?." 2007 International Joint Conference on Neural Networks. IEEE, 2007.
>
> [2] Hanika, Tom, Friedrich Martin Schneider, and Gerd Stumme. "Intrinsic dimension of geometric data sets." Tohoku Mathematical Journal 74.1 (2022): 23-52.

---

> > ### Author Response · Authors · 2022-11-15
> > **Minor Remarks**
> >
> > We thank the reviewer for reading carefully and spotting typos and details.
> >
> > > p3, top: the choice of a measure $\mu$ on $\mathbb{R}$ is confusing, it was a measure on $X$ two paragraphs earlier. Also it is confused with $\nu$ after
> >
> > We do not have a measure on $\mathbb{R}$ here, but the counting measure on the set $\{\emptyset\}$. In this dataset, $\mathbb{R}$ is the feature set. To be more specific, we here identify every real number $r \in \mathbb{R}$ with the map $\{ \emptyset \} \to \mathbb{R}, \emptyset \mapsto r$. In general, we use $\mu$ for measures and $\nu$ for the more specific case of normalized counting measures.
> >
> > >the notation $-\alpha$(instead of just $\alpha$ ) is weird, is it motivated in earlier work?
> >
> > Yes, we adapted this notion from earlier work.
> >
> > >the "without loss of generality" in the proof of Lemma 3.1 is not so trivial, maybe it is possible to be more rigorous here
> >
> > The without loss of generality here comes from the fact that if there is an $x$ breaking the condition, we can simply add it to $M$. This will not change $z$.
> >
> > >unless I am mistaken, in Lemma 3.4, $\phi$ is the minimum of the set, not simply a member of. It is computed that way in the algorithm.
> >
> > Thank you for carefully reading our statements and bringing this up. We modified the statement and the proof in the revised version.
> >
> > >-the notation ... is confusing, isn't it just ... times ...?
> >
> > Yes, it is. We used this notation to emphasize that all summands $\phi_j$ are replaced by $\phi_{s_i}$ or $\phi_{s_{i+1}}$. However, if too confusing, we are willing to change it for the camera ready.
> >
> > > I think theorem 4.1 is immediate[...]
> >
> > We agree that the statement is very intuitive. We still would like to include the formal proof for the sake of completeness.
> >
> > >  - It is quite counter-intuitive that the saved computation cost is not strictly decreasing with the size of the support sequence, it needs to be written and emphasized
> >
> > It is true, that computing the $\phi_j$ for the "gap" elements between the points of the support sequences becomes cheaper with more elements in the support sequence. However, the costs of computing $\phi_{s_i}$ for the elements of the support element themselves rise. Hence, there is a sweet spot which balances the increasing costs for the support sequence and the decreasing costs for the "gaps". We mention this in the revised version.
> >
> > >Maybe the table 2 of results could be better presented as curves, that will better outline the potential correlation between ID and performance ?
> >
> > Thanks for that suggestion. We will try it out for the camera ready version.

---

### Review · Reviewer_Tgwr · 2022-11-03

**Summary Of Contributions:**

This paper considers an obscure notion of dimensionality (proposed by V. Pestov 20 years ago), and proposes a heuristic to speed up its computation with some error.  Experiments show the error of the heuristic is small on some data sets realized by graphs.


**Audience:**

No

**Claims And Evidence:**

Yes

**Requested Changes:**

This paper would need significant changes for me to recommend its acceptance at TMLR.
 - provide intuition for why this notion of ID is capturing dimension
 - provide motivation for why this method is useful for machine learning, and why it is superior to other notions in the literature.  I expect this would require a completely different experimental section.
 - improve the writing to make the description of Algorithm 2 more clear (just pseudo code and a few comments its not sufficient
 - improve writing to explain what k-hop graphs are, and why the ID values computed measure a useful notion of dimensionality for those graphs, and why those associated numerical values are related and important for machine learning
 - explain the relationship between l  (the size of the subset) and the approximation that can be achieved (its ok if some guarantee does not match the very small error found on some data sets, but you want to know how large it could be on *any* data set).

**Strengths And Weaknesses:**

The methods are limited to data where distance is measured with respect to a feature set F, and one computes a distance d_F(a,b) = max_{f in F} |f(a) - f(b)|, and |F| is a finite set.  The algorithm iterates over this set, its runtime is linear in |F|.  I interpret the |F| as the original dimension, but note that this notion is not powerful enough to capture the common Euclidian l_2 distance, where the corresponding "test" function is a unit vector u in R^d, and f_u(a) = <u,a>.  Then one can write
  ||a-b||_2 = sup_{f_u in F} |f_u(a) - f_u(b)| = sum_{u a unit vector} |<u,a> - <u,b>|.
Note that in this case, when in greater than 1 dimension, then F is not finite, so this method does not apply.

This would apply for instance with the l-infinity distance on vector data.
The paper mostly finds applications in analyzing large "k-hop" graphs where the finite set F.  As someone unfamiliar with this definition (of k-hop graphs) before the paper, I was not able to fully understand it or its relevance from the description in the paper (start of Section 5 and Figure 1).

In short, the studied notion of intrinsic dimensionality (ID) of a data set X is 1/Delta(X)^2 where Delta(X) is the average "partial diameter" of X under d_F, where partial diameter is over the most compact subset of a 1-alpha fraction of the measure of X, averaged over all alpha in (0,1).
While the paper mentions ways this notion of ID might be useful, it does not provide motivation for it, does not explain how it has a modeling advantage over more standard (Euclidean) distance based notions, and does not demonstrate its use in any application.
This concept was new to me as a reader, and I did not come away with a sense of why it captured dimensionality, or why this notion was useful.  So did not find a strong motivation in machine learning for this study from the writing in the paper.


The main contribution of the paper is to provide a heuristic for computing a quantity that should be close to the intended ID value on a finite point set X (size n) with a finite set of feature functions F.  The key is to compute the partial diameter for X for each f in F and each alpha in (1/n, 2/n, ..., (n-1)/n).  The central idea in this paper is to find a small S subset X or size l, and for each f in F, compute the partial diameter values on S instead of X.  Through a simple argument, it provides quantities that are under- and over-estimates of the true values.  On experiments, it shows these upper and lower bounds are close to each other on the graph data sets considered.

The paper does not prove a relationship between l (the size of the subset S), and the amount of error.  Hence it is not an approximation algorithm, but only a heuristic.


Ultimately,
 - I did not find the problem well-motivated
 - I did not find the algorithmic insights that deep
 - I found the writing hard to follow (e.g., the new method, Algorithm 2 was hard to follow)
 - The paper did not convince me the proposed approach was useful in machine learning.
So I do not recommend acceptance at TMLR with significant changes.

---

> ### Author Response · Authors · 2022-11-15
> **Response to Review**
>
> Dear Reviewer,
> we thank you for your detailed review and the suggestions. We hope that we can clarify all remarks with the following comments and the revised version.
>
> >This paper considers an obscure notion of dimensionality (proposed by V. Pestov 20 years ago)[..]
>
> We agree that the Concept from Pestov is different that estimating manifold dimensions. However, we respectfully disagree that it is obscure. Several fundamental papers have been published in established AI venues [1][2]. Furthermore, they have led to insights about common machine learning topics as nearest neighbor classifiers[3] and similarity search[4].
>
> Furthermore, concentration phenomenons are of high interest in basic mathematical research[5-7] and it has been shown, that they are present in many settings. Thus, it is fruitful and imperative to build bridges to machine learning.
> Consequently, concentration phenomena are recently considered in the realm of machine learning and computer science [8-10].
>
>
> [1] Pestov, Vladimir. "Intrinsic dimension of a dataset: what properties does one expect?." 2007 International Joint Conference on Neural Networks. IEEE, 2007.
>
> [2] Pestov, Vladimir. "An axiomatic approach to intrinsic dimension of a dataset." Neural Networks 21.2-3 (2008): 204-213.
>
> [3] Pestov, Vladimir. "Is the k-NN classifier in high dimensions affected by the curse of dimensionality?." Computers & Mathematics with Applications 65.10 (2013): 1427-1437.
>
> [4]  Pestov, Vladimir. "Indexability, concentration, and VC theory." Proceedings of the Third International Conference on SImilarity Search and APplications. 2010.
>
> [5] Kazukawa, Daisuke. "Concentration of product spaces." Analysis and Geometry in Metric Spaces 9.1 (2021): 186-218.
>
> [6] Eldan, Ronen, and Renan Gross. "Concentration on the Boolean hypercube via pathwise stochastic analysis." Proceedings of the 52nd Annual ACM SIGACT Symposium on Theory of Computing. 2020.
>
> [7] Funano, Kei. "Concentration of 1-Lipschitz maps into an infinite dimensional ℓ^{𝑝}-ball with the ℓ^{𝑞}-distance function." Proceedings of the American Mathematical Society 137.7 (2009): 2407-2417.
>
> [8] Bac, Jonathan, and Andrei Zinovyev. "Local intrinsic dimensionality estimators based on concentration of measure." 2020 International Joint Conference on Neural Networks (IJCNN). IEEE, 2020.
>
> [9] Raginsky, Maxim, and Igal Sason. "Concentration of measure inequalities in information theory, communications, and coding." Foundations and Trends® in Communications and Information Theory 10.1-2 (2013): 1-246.
>
> [10] P. L. Bartlett and S. Dasgupta, Topics in statistical learning theory. Paris: Société Mathématique de France (SMF) (2022; Zbl 07564348)
>
> >  The methods are limited to data where distance is measured with respect to a feature set F [... I interpret the |F| as the original dimension.
>
>  It is true that $F$ reflects the "original dimension" when we choose F as the set of coordinate projections. However, F is NOT restricted to that choice. It can be used to incorporate different information of the data into the dimension computation. More generally, the feature set $F$ can be comprised of any set of real-valued function fulfilling the criteria given in the paper. In our experiments, for example, we incorporate graph structures into the computation of the intrinsic dimension. Furthermore, in [1] $F$ is used to model distances (including Euclidean distances!) for distance-based learning or to give an ID notion to incidence structures.
>
>    To sum up, the modeling via a feature set $F$ is not a limitation. Instead, as indicated above, it allows to incorporate different kinds of information into the computation of the intrinsic dimension.
>
> [1] Hanika, Tom, Friedrich Martin Schneider, and Gerd Stumme. "Intrinsic dimension of geometric data sets." Tohoku Mathematical Journal 74.1 (2022): 23-52.
>
> >  The paper mostly finds applications in analyzing large "k-hop" graphs[..]  I was not able to fully understand it or its relevance[..]
>
> In this work, we do not consider k-hop graphs (this term is not present in our paper). We consider graph datasets, which consist of a graph and node attribute vectors which reflects the basic setting for OGB node property prediction tasks. For such data and a given parameter $k$ we define the $k-$hop geometric dataset, that consists of the following features:
>   - The mapping to the individual attribute values, i.e., the different columns of X
>   - These mappings are composed with the graph structure by multiplying X with $\tilde{A}$ for $i \leq k$ times.
>
> This geometric dataset is motivated by a class of GNNS which lay the foundation of many well performing methods for OGBN. As mentioned in our paper, these methods use networks that employ $X,\tilde{A}X, \dots, \tilde{A}^k X$ as input. This corresponds our definition of the $k-$hop features. We extend this motivation in our revised version.

---

> > ### Author Response · Authors · 2022-11-15
> > **Reviewer Response Part 2**
> >
> > >   In short, the studied notion of intrinsic dimensionality (ID) [.. I did not come away with a sense of why it captured dimensionality, or why this notion was useful. So did not find a strong motivation in machine learning for this study from the writing in the paper.
> >
> > Pestov defines the Curse of Dimensionality as "[...] a name given to the situation
> > where all or some of the important features of a dataset
> > sharply concentrate near their median (or mean) values and thus
> > become non-discriminating. In such cases, X is perceived as
> > intrinsically high-dimensional." [1] Thus, the ID estimator used aims to compute to which extent the features allow to discrimintate different data points. For a specific feature f \in F, [2] therefore defines the Partial Diameter of $f$ with regard to a specific $\alpha$ such that it displays to which extent $f$ can subsets of the specific minimal measure $1-\alpha$. The observable Diameter with respect to $\alpha$ then defines to which extent $F$ can discriminate points of the minimum measure 1-$\alpha$ by being defined as the supremum of the Partial diameter of all $f \in $F. To observe the discriminability for different minimal measures $\alpha$, we then build $\Delta(\mathcal{D})$ by integrating over $\alpha$ in the range from zero to one. Thus, higher values of $\Delta\mathcal{D}$ then indicate a high discriminability of the data points. Thus, they should be of lower intrinsic dimensionality with respect to Pestovs definition above. Furthermore, the ID $\partial (\mathcal{D}) = \frac{1}{\Delta(\mathcal{D})}²$ respects the axiomatization that [2] proposed, which we added to the revised version.
> >
> > To sum up, the proposed ID measure do NOT approximate the dimensionality of a manifold in the euclidean sense, but directly reflects to which extent a geometric data set is effected by the curse of dimensionality, which is defined as above..
> >
> > Note, that we found that the first step of neighborhood aggregation was always crucial for classification performance. Furthermore, it always led to the highest drop in ID with respect to our estimator. This indicates, that the discrimnability has a connection to classification performance. This drop in the ID is not captured by MLE.
> >
> > > The paper does not prove a relationship between l (the size of the subset S), and the amount of error. Hence it is not an approximation algorithm, but only a heuristic.
> >
> >  It is true that we do not have an error bound that is independent from the approximation itself. Note, that such a bound would not only depend on $l$ but also on the "spacing" of the support elements. However, our method still is an approximation and not just a heuristic as Corollary 4.5 gives us control over the error and an approximation bound which just depends on the approximations themselves. The computation of the exact ID is NOT needed. Furthermore, as brought up by Reviewer pGDm, if an approximation is not "accurate enough", a finer approximation can be computed from it with the addition of further support elements.  Furthermore, note, that in most cases (all considered datasets except ogbn-papers100M) it is possible to compute the exact ID from the support sequence. Hence, in such cases, the intrinsic dimensionality does not depend any more on s. The choice of $s$ is only relevant for the computation steps, not the result itself.
> >
> > To underpin the above we extended this part in our revised version. To gain further insights in the approximation behavior, we conducted additional experiments on random data consisting of 1 million, 10 million and 100 million data points with 10, 50 or 250 attributes. In all cases, we have an accuracy of over 99,9% with a geometric support sequence with l=100,000.
> >
> > To sum up,  our approximation is useful as we give an error bound which can be computed without having the exact ID. Furthermore, we now have  empirical results both on real world graph data and random data that shows that support sequences with $l=100,000$ lead to accuracies of over 99,9%.
> >
> > We added further random experiments to the revised version.
> >
> > [1] Pestov, Vladimir. "Intrinsic dimension of a dataset: what properties does one expect?." 2007 International Joint Conference on Neural Networks. IEEE, 2007.
> >
> > [2] Hanika, Tom, Friedrich Martin Schneider, and Gerd Stumme. "Intrinsic dimension of geometric data sets." Tohoku Mathematical Journal 74.1 (2022): 23-52.

---

> > > ### Author Response · Authors · 2022-11-15
> > > **Reviewer Response Part 3**
> > >
> > > > I did not find the problem well-motivated
> > >
> > > Please see our motivation and explanation above.
> > >
> > > > I did not find the algorithmic insights that deep
> > >
> > > The deepness solely arises by the theoretical results. We agree, that no further tricks or insights result from Algorithm 2. However, using our Lemmata, it allows to compute a phenomenon that was before mostly theoretically observed or computed on moderate-sized data on data sets with 100 of million of nodes. We believe, that this makes Algorithm 2 interesting and important.
> > >
> > > > I found the writing hard to follow (e.g., the new method, Algorithm 2 was hard to follow)
> > >
> > > We added further explanation to the revised version with more details on what the Lines in the Algorithm are for.
> > >
> > > >The paper did not convince me the proposed approach was useful in machine learning.
> > >
> > > Please see again our motivation above and the cited sources for measure concentration  investigated in Computer Science and Mathematics.
> > >
> > > > provide intuition for why this notion of ID is capturing dimension
> > >
> > > We again refer to our explanation above.
> > >
> > >
> > > >  improve writing to explain what k-hop graphs are, and why the ID values computed measure a useful notion of dimensionality for those graphs, and why those associated numerical values are related and important for machine learning
> > >
> > > Please again see our answer above.
> > >
> > > >  explain the relationship between l (the size of the subset) and the approximation that can be achieved (its ok if some guarantee does not match the very small error found on some data sets, but you want to know how large it could be on any data set).
> > >
> > > Please see our arguments above regarding this point.

---

> > > > ### Comment · Reviewer_Tgwr · 2022-11-20
> > > > **Response to comments**
> > > >
> > > > Dear authors:  Thank you for your revisions and explanations.  I think the paper has improved.
> > > >
> > > >  - I now have a better idea of the connection to machine learning.  I found and read some of Pestov's paper, and via their extended discussions on connections to tasks within data analysis, I have a better sense of the how this notion of ID is supposed to be useful.  It is a curious perspective, but to be honest, I am not personally convinced it is useful.  But I guess the resolution of that question is not the point of this particular paper.
> > > >
> > > >  - I also noticed several papers of those mentioned in the comments identified the (estimated) computation of this form of ID as an important challenge.
> > > >
> > > >  - While I am still not convinced about useful this paper's characterization of **when** its method works, but it is more clear now on more examples that it is are converging.
> > > >
> > > >  - Out of curiosity, for the new non-graph example in Section 6, how is $\nu$ chosen, and what value of ID was determined for these data sets?  How does it change with the ambient dimension d?

---

> > > > > ### Author Response · Authors · 2022-11-25
> > > > > **Answer on Questions in the Response**
> > > > >
> > > > > Dear Reviewer Tgwr,
> > > > >
> > > > > thanks for the quick response. We are glad about your appreciation of our effort. Furthermore, we thank you for acknowledging that the literatur identifies estimating concentration based IDs as an important challenge.
> > > > >
> > > > > Concerning your questions, $\nu$ is, as in all of our experiments, the normalized counting measure. We observe that the ID is nearly independent of the ambient dimension d. This observation can be attributed to our procedure of generating the random data, i.e., we drew normal distributed values with mean of 0 and different values for the standard deviation $\sigma$ with the random.normal function of numpy. However, $\Delta(\mathcal{D})$ scales linearly with the standard deviation from which we drew the random values. This reflects that our notion for ID does account for the actual discriminability of the data, not the ambient dimension.
> > > > >
> > > > > This is in particular interesting if we compare random data to one-hot encoded data. To be more specific, the $n$-dimensional unit vectors have a discriminability of $\Delta(\mathcal{D})=\frac{n-1}{n}$. Hence, $\partial (\mathcal{D}) \rightarrow 1$ for $n \rightarrow \infty$.

---

### Review · Reviewer_HCNE · 2022-11-05

**Summary Of Contributions:**

This paper looks at the Intrinsic Dimension (ID) estimator from Hanika, Schneider, and Stumme 2022. The paper presents two algorithms, the first algorithm is a quadratic algorithm for computing this ID estimate for finite data sets and the second algorithm is faster for approximating the same ID estimate.

The paper has a few proofs to justify their algorithm. They then test the approximation algorithm to determine the error rate and time saved.

The paper then computes this ID for a variety of different datasets.

**Audience:**

Yes

**Claims And Evidence:**

Yes

**Requested Changes:**

I would like the authors to at least address the first weakness and better explain the motivation for this estimate and the experiments in section 5.2

**Strengths And Weaknesses:**

**Strengths**


The algorithm presented is simple and easily implementable and has rigorous proofs

**Weaknesses**

1) The motivation for computing this particular ID estimate is not clear to me. In relation to this I don't the goal of the experiment in section 5.2. As we can see the two ID estimates can be very very different (eg. 3 vs over 1 million).
2) While most of the proofs are clear. I think the proofs could be further simplified (but this is minor complaint)

**Questions**


1) Why does the saved time initially increase? Should't the curve be monotonic?
2) Do the authors have any intuition as to why the dimension should drop as we increase $k$?

---

> ### Author Response · Authors · 2022-11-15
> **Response to Review**
>
> We thank the Reviewer for the suggestions. They definitely enhance our paper.
>
> > The motivation for computing this particular ID estimate is not clear to me. In relation to this I don't the goal of the experiment in section 5.2. As we can see the two ID estimates can be very very different (eg. 3 vs over 1 million).
>
> Pestov defines the curse of dimensionality as "[...] a name given to the situation
> where all or some of the important features of a dataset
> sharply concentrate near their median (or mean) values and thus
> become non-discriminating. In such cases, X is perceived as
> intrinsically high-dimensional." [1] Thus, the ID estimator aims to compute to which extent the features allow to discriminate different data points. For a specific feature f \in F, [2] therefore defines the Partial Diameter of $f$ with regard to a specific $\alpha$ such that it displays to which extent $f$ can discriminate subsets with minimal measure $1-\alpha$. The observable diameter with respect to $\alpha$ then defines to which extent $F$ can discriminate points with minimum measure 1-$\alpha$ by being defined as the supremum of the partial diameter of all $f \in F$. To observe the discriminability for different minimal measures $\alpha$, we define $\Delta(\mathcal{D})$ by integrating over $\alpha$ in the range from zero to one. Thus, higher values of $\Delta\mathcal{D}$ indicate a high discriminability and hence a lower intrinsic dimensionality with respect to Pestovs definition above. Furthermore, the ID $\partial (\mathcal{D}) = \frac{1}{\Delta(\mathcal{D})}²$ respects the formal axiomatization [2] for ID functions, which we added to the paper.
>
> To sum up, the proposed ID measure does NOT approximate the dimensionality of a manifold in the Euclidean sense, but directly reflects to which extent a geometric data set is effected by the curse of dimensionality.
>
> This extended explanation and a high level description of the different axioms has been added to the paper.
>
> [1] Pestov, Vladimir. "Intrinsic dimension of a dataset: what properties does one expect?." 2007 International Joint Conference on Neural Networks. IEEE, 2007.
>
> [2] Hanika, Tom, Friedrich Martin Schneider, and Gerd Stumme. "Intrinsic dimension of geometric data sets." Tohoku Mathematical Journal 74.1 (2022): 23-52.
>
> Regarding the experiments:
> It is true that the values of intrinsic dimensionality via our estimator and MLE are not comparable in absolute terms, since, as explained above, they aim to capture different phenomena. The important observation from the experiments is, that the first step of neighborhood aggregation is the most crucial one for enhancing classification performance and also the one that often leads to a huge drop in ID with respect to our estimator. This drop indicates a better discriminability of data points. However, this drop is not captured by MLE. To get more insights into the used ID notion, we have added further experiments on random data and on one-hot encoded graphs with rising edge sizes. Please see the revised version for more details.
>
> >While most of the proofs are clear[...]
>
> We agree that parts of the proofs are straight-forward and that the statements are intuitive. Still, we would like to keep the proofs in their comprehensiveness for the sake of comprehensibility and reproducibility.
>
> >Why does the saved time initially increase? Should't the curve be monotonic?
>
> With longer support sequences, the costs of computing $\phi_k$ for the $k$s between the support elements drop. However, the costs for computing the $\phi$ values for the elements of the support sequence itself rise with longer support sequence. Hence, there is a sweet spot between the costs raised by the support sequence and the reduced costs of the remaining calculations.
>
> >Do the authors have any intuition as to why the dimension should drop as we increase ?
>
> Our ID estimator expresses to which extent we are able to distinguish data points. Thus, adding additional features via neighborhood aggregation adds further useful information that can help to separate different data points. This results in a decreasing ID. Note, that the step from $k=0$ to $k=1$ is the first one that introduces features based on the graph structure and thus allows to use neighborhood information for discriminating data points. This might be reason why this step led to the biggest ID drop in our experiments.

---

> > ### Comment · Reviewer_HCNE · 2022-11-19
> > **Thank you for the clarification**
> >
> > Those seem like both non - standard definitions of curse of dimensionality and hence of intrinsic dimension. Hence I think this discussion should be included in the paper.
> >
> > That being said, the degree of indistinguishability of data is an important concept. That makes the motivation a lot clearer.

---

> > > ### Author Response · Authors · 2022-11-19
> > > **Motivation**
> > >
> > > Dear Reviewer HCNE,
> > > thank you for the quick response. We are happy that our explanation above helped to clarify the motivation. The explanation is part of the revision which is the actual version here on Open Review.
> > > Kind Regards,
> > > The authors

---

### Author Response · Authors · 2022-11-15
**Revised Version and Answer to Reviews**

Dear Reviewers and Action-Editor,

thank you for the detailed reviews and the helpful suggestions. They helped to further enhance our paper. With the incorporation of the additional ideas and requests raised by the reviewers, the main content has now grown to 14 pages. We therefore changed our submission to a "Long submission".

Thank you all for your kind efforts.

Best regards,

The Authors

---

### Author Response · Authors · 2023-01-11
**Camera Ready Version**

Dear Reviewers and Editors,

we just uploaded the camera ready version. Thank you for the extensive feedback and the productive discussion period. Submitting to TMLR was a pleasant experience for us.

Best regards,

The authors

---

### Decision · Action_Editors · 2022-12-26

**Recommendation:** Accept as is

**Comment:**

The paper considers a non-standard notion of intrinsic dimension based on the concentration-of-measure, and provides computationally efficient algorithms to approximate this notion of dimension. The proposed methods are then tested on large data sets.

The reviewers all pointed out the lack of motivation of this non-geometrical notion of intrinsic dimension. But the discussion period has been fruitful and has lead to major improvement of the paper, in particular on this aspect. Additional experiment have been added, too.

Despite the reserve of one reviewer concerning the usefulness of the notion for ML, the claims are well supported and thus the paper deserves publication. Indeed, TMLR guidelines mention that the two main criteria for acceptance are:
"Are the claims made in the submission supported by accurate, convincing and clear evidence?" It the case here.
"Would some individuals in TMLR's audience be interested in the findings of this paper?" I believe so, even if it may a bit niche.

Given all this I recommend acceptance as such.



**Audience:**

The audience is appropriate (despite one reviewers has some reservations on the usefulness of the proposed ideas to machine learning). The results will mostly interest already specialists of the field of intrinsic dimension, and potential practitioners who are keen to test novel ID approximation algorithms for very large data sets.

**Claims And Evidence:**

As pointed by all reviewers, the paper contains enough evidence to support the claims: both theoretical results with associated proofs (which are clear), and numerical experiments on known data sets and artificial data too.